



# SedTrace 1.0: a Julia-based framework for generating and running reactive-transport models of marine sediment diagenesis specializing in trace elements and isotopes

Jianghui Du[1]

[1]Institute of Geochemistry and Petrology, Department of Earth Sciences, ETH Zürich, Zürich, 8092, Switzerland

*Correspondence to*: Jianghui Du (jianghui.du@erdw.ethz.ch)

**Abstract.**

Trace elements and isotopes (TEIs) are important tools in studying ocean biogeochemistry. Understanding their modern ocean budgets and using their sedimentary records to reconstruct paleoceanographic conditions require mechanistic understanding of the diagenesis of TEIs, yet the lack of appropriate modeling tools has limited our ability to perform such research. Here we introduce SedTrace, a modeling framework that can be used to generate reactive-transport code for modeling marine sediment diagenesis and assist model simulation using advanced numerical tools in Julia. SedTrace enables mechanistic TEI modeling by providing flexible tools of pH and speciation modeling, which are essential in studying TEI diagenesis. SedTrace is designed to solve one particular challenge facing the users of diagenetic models: existing models are usually case-specific and not easily adaptable for new problems, such that the user has to choose between modifying published code and writing their own code, both of which demand strong coding skills. To lower this barrier, SedTrace can generate diagenetic models only requiring the user to supply Excel spreadsheets containing necessary model information. The resulting code is clearly structured and readable, and is integrated with Julia's differential equation solving ecosystems, utilizing tools such as automatic differentiation, sparse numerical methods, Newton-Krylov solvers and preconditioner. This allows efficient solution of large systems of stiff diagenetic equations. We demonstrate the capacity of SedTrace using case studies of modeling the diagenesis of pH, radiogenic and stable isotopes of TEIs.

## 1 Introduction

Trace elements and isotopes (TEIs) play critical roles in the heathy functioning of the marine ecosystem (SCOR Working Group, 2007). For example, Fe, Zn, Ni and other transition metals serve as cofactors in metalloenzymes that are essential for phytoplankton physiology, such that the low concentrations of these metals can limit phytoplankton productivity with implications for the global carbon cycle (Martin et al., 1987; Vance et al., 2017; Tagliabue et al., 2017; Morel et al., 2020; Lemaitre et al., 2022). TEIs are also powerful tracers of geological, physical and biogeochemical processes in the ocean (Lam and Anderson, 2018). For instance, the radiogenic isotope of Nd has been used to study ocean circulation, and the role of continental weathering and tectonics in past climate changes (Piepgras et al., 1979; Palmer and Elderfield, 1985; Frank, 2002;





Goldstein and Hemming, 2003; Du et al., 2020); the stable isotopes of Mo have been used to study past ocean oxygenation (Archer and Vance, 2008; Anbar and Rouxel, 2007; Lyons et al., 2009). Since the dawn of the international GEOTRACES program, there have been increasing efforts aimed at characterizing the distributions, global oceanic budgets, internal cycling and external sources of TEIs in the modern ocean and their utility in paleoceanography (Anderson, 2020; Schlitzer et al., 2018).

       In recently years, however, two issues have emerged as major challenges to the understanding and application of
marine TEIs. First, the modern oceanic budgets of many TEIs, such as Fe, Cu, Ni, Zn and Nd, cannot be balanced by known sources at the riverine, atmospheric and hydrothermal interfaces, and increasing evidence suggests fluxes across the sediment-water interface (SWI) may play important, if not dominant, roles in setting the global oceanic budgets of TEIs (Homoky et al., 2016; Jeandel, 2016; Little et al., 2014; Du et al., 2020; Haley et al., 2017; Elrod et al., 2004; Cameron and Vance, 2014; Little et al., 2020). Second, applying TEIs as proxies to study the geological evolution of the ocean system is hampered by the lack
of quantitative knowledge of the sedimentary cycling of TEIs, which may alter or completely erase the original proxy signals preserved in sedimentary archives (Horner et al., 2021; Crusius and Thomson, 2000). Models of the sedimentary cycling of TEIs are thus needed to resolve these issues. Quantitative modeling is particularly indispensable in this case, because the complexity, heterogeneity, and highly coupled nature of biogeochemical and physical processes in marine sediments make straightforward interpretation of measured modern and paleo TEI data difficult. In addition, TEI data in sediment pore water
is extremely scarce because of analytical and sampling challenges. We thus hope that models can assist our understanding of the sedimentary TEI cycling in the presence of large data gaps.

       A diagenetic model is typically a 1-D reactive-transport model that includes physical transport and biogeochemical reactions to simulate the distributions of dissolved substances in pore water and solid substances in sediments (Berner, 1980; Boudreau, 1997; Burdige, 2006). Such models have traditionally been used to study the sedimentary cycling of carbon, oxygen
and nutrients (Wang and Van Cappellen, 1996; Boudreau, 1996; Meysman et al., 2003; Soetaert et al., 1996), and to a limited extent TEIs like Fe, Mn and U (Dale et al., 2015; Burdige, 1993; Lau et al., 2020; Maher et al., 2006). However, to date there has been little concerted effort to develop diagenetic models specifically for TEIs, due to some particular challenges of TEI biogeochemistry. First, a realistic digenetic TEI model needs to include a pH module to enable speciation modeling. Pore water pH is difficult to model because of the large network of biogeochemical and physical processes involved (Boudreau and
Canfield, 1988, 1993; Jourabchi et al., 2008; Reimers et al., 1996). Many studies thus ignore TEI speciation in diagenetic models. Moreover, TEIs are commonly influenced by more biogeochemical reactions than are the major and minor constituents. Together with the necessity for pH and speciation modeling, a diagenetic model may need to include a large system of differential equations that are also highly stiff because the span of the reaction timescales can be large (Boudreau, 1997). Thus, unlike traditional diagenetic models of carbon, oxygen and nutrients, diagenetic TEI models need also to take
into account numerical efficiency, especially if the goal is to couple diagenetic models to global ocean biogeochemical models (Hülse et al., 2018; Archer et al., 2002).

       The greatest challenge from the user's point of view is that most diagenetic models are too specialized to be adaptable. The strong heterogeneity of marine sediments implies that no single diagenetic model can be created to include all sedimentary



processes applicable to all environmental settings. Often, modelers create specific diagenetic models with fixed selection of substances and processes for their own studies. The model code is often not open source, and it could be time-consuming and error-prone for other users to adapt the code for new studies. Writing and modifying code require strong programming skills, which is a significant hurdle to the general user. Thus, rather than creating one all-encompassing diagenetic model, it is preferable to create a modeling framework that can generate custom models, allowing users with limited programming skills to create and run models satisfying their own needs (Soetaert and Meysman, 2012).

In this study, we describe SedTrace 1.0, a modeling framework that automates the generation of Julia code for diagenetic models and provides high-performance computing tools to assist model simulation, only requiring the user to supply information using a Microsoft Excel input file. SedTrace specializes in modeling TEIs, and can accommodate a wide range of custom pH and speciation modeling choices. The design principle is to give as much control to the user as possible when generating the model, such that SedTrace is only meant to help converting user ideas to code rather than making model decisions for the user. Julia is an open source, dynamically typed programming language for high-performance scientific computing (Bezanson et al., 2017). It offers the high productivity of scripting languages like Python, but can also match the performance of statically typed languages such as C and FORTRAN. It is increasingly being adopted by the climate and ocean modeling community (Pasquier et al., 2022; Sridhar et al., 2022; Sulpis et al., 2022), with well-supported ecosystems relevant to solving differential equations (Rackauckas and Nie, 2017).

This paper is structured as following. First we describe the model equations and framework in Sect. 2. We then discuss the code generation procedure for physical processes and biogeochemical reactions in Sect. 3 to 5. Numerical solution of the model is discussed in Sect. 6 and 7. Finally, we present a few case studies in Sect. 8 and briefly touch on future development in Sect. 9.

## 2 Model framework

SedTrace uses the 1-D diagenetic equation (Boudreau, 1997; Meysman et al., 2003):

$$\frac{\partial}{\partial t}\phi^{\xi}C_i^{\xi} + T_i = \phi^{\xi}R_i, \tag{1}$$

$$T_i = \frac{\partial}{\partial x}F_i^{\xi,adv} + \frac{\partial}{\partial x}F_i^{\xi,diff} + \phi^{\xi}T_i^{bio}, \tag{2}$$

where $C_i^{\xi}$ is the concentration of model substance $i$ in the phase $\xi$ ($f$ for pore water or $s$ for solid sediment), $\phi^{\xi}$ is the phase volume fraction, $T_i$ is the transport due to advection, diffusion and bio-irrigation, $R_i$ is the net source or sink due to biogeochemical reactions, $t$ is time and $x$ is the sediment depth-coordinate staring from the SWI pointing downward. In SedTrace, the default units are "mmol" for mass, "year" for time, "cm" for length. For example, reaction rates are in units of "mmol cm$^{-3}$ yr$^{-1}$".

The transport terms have well-established forms in diagenetic models (Berner, 1980; Boudreau, 1997). It is largely the reaction terms which are case-specific that create the diversity of diagenetic models and thus cause their poor adaptability.





SedTrace fixes the general forms of the transport terms and only requires the users to supply case-specific parameters, while letting the user to set the reaction terms freely.

Using user specified spatial grids, which can be non-uniform, SedTrace discretizes Eq. (1) using the method of lines (Boudreau, 1996), resulting in a system of ordinary different equations (ODEs) of time only:

$$\frac{d}{dt}C_{i,j}^{\xi} = \frac{F_{i,j-\frac{1}{2}}^{\xi,adv}-F_{i,j+\frac{1}{2}}^{\xi,adv}}{\phi^{\xi}\Delta x_j} + \frac{F_{i,j-\frac{1}{2}}^{\xi,diff}-F_{i,j+\frac{1}{2}}^{\xi,diff}}{\phi^{\xi}\Delta x_j} + R_{i,j} - T_{i,j}^{bio},$$  (3)

where $j$ is the index of the grid cell. $F_{i,j-\frac{1}{2}}^{\xi}$ and $F_{i,j+\frac{1}{2}}^{\xi}$ are the fluxes across the left and right boundaries of the cell respectively. $\Delta x_j$ is the volume of the cell.

The diffusive and advective fluxes are discretized using the Finite Volume-Complete Flux (FV-CF) scheme (Llorente et al., 2020; Boonkkamp and Anthonissen, 2010; ten Thije Boonkkamp and Anthonissen, 2011). The finite volume flux $F_{i,j+\frac{1}{2}}^{\xi}$ is derived based on the analytical solution of the local two-point boundary value problems, and is partitioned into a

homogeneous flux and an inhomogeneous flux. Presently SedTrace only uses the homogeneous flux term. The symbolic derivation of these fluxes using *Mathematica* can be found in /math/Flux.nb. The resulting discretization maybe viewed as a weighted average of the centered scheme and the upwind scheme, and the weight is controlled by the cell Péclet number $\frac{v^{\xi}\Delta x}{2D^{\xi}}$, where $v^{\xi}$ is the advective velocity and $D^{\xi}$ is the diffusion coefficient. At high Péclet numbers (advection dominant), the scheme approaches the upwind scheme, while at low Péclet numbers (diffusion dominant) it approximates the

centered scheme. This ensures at least first order uniform convergence regardless of the Péclet number. Similar Péclet-number-dependent schemes have been applied to diagenetic and ocean modeling (Fiadeiro and Veronis, 1977; Soetaert and Meysman, 2012; Boudreau, 1996).

SedTrace collects the discretized model substances into an $MN$ vector:

$$\mathbf{C} = \left[C_{1,1}, C_{2,1}, ..., C_{M,1}, ..., C_{i,j}, ..., C_{1,N}, C_{2,N}, ...C_{M,N}\right],$$  (4)

where $M$ is the number of model substances and $N$ is the number of grid cells. The system of ODEs including boundary conditions can be written in the matrix form:

$$\frac{d}{dt}\mathbf{C} = \mathbf{AC} + \mathbf{BC} + \mathbf{b} + \mathbf{S}.$$  (5)

$\mathbf{A}$ is an $MN \times MN$ matrix including the linear diffusion and advection operators. $\mathbf{B}$ is an $MN \times MN$ matrix containing the homogeneous part of the boundary conditions. $\mathbf{b}$ is an $MN$ vector containing the non-homogeneous part of the

boundary conditions. $\mathbf{S}$ is an $MN$ vector, generally a nonlinear function of $\mathbf{C}$, incorporating the reaction and bio-irrigation sources and sinks, and nonlinear coupling of transport.

To generate code for Eq. (5) the user supplies an Excel file model_config.xlsx to SedTrace. An example of a simple Fe cycle model (SimpleFe) can be found in /examples/SimpleFe/model_config_simpleFe.xlsx, and the spreadsheets are shown in Table 1 to 7. The substances sheet (Table 1) lists the modeled substances, their types (e.g.,

solid or dissolved), chemical formula and boundary conditions. The reactions sheet (Table 2) lists the kinetic reactions,





their chemical equations and rate expressions. The `speciation` sheet (Table 3) lists aqueous speciation reactions. The `adsorption` sheet (Table 4) lists the adsorbed species. The `diffusion` sheet (Table 5) lists information to compute the diffusion coefficients of dissolved substances. The `parameters` sheet (Table 6) lists the parameters required by the model. The `output` sheet (Table 7) is used to formulate output and plotting. The `data` sheet (not shown here) includes observational

data that will be plotted together with model outputs.

The workflow of generating and running diagenetic models using SedTrace is shown in Fig. 1. In Sect. 3 and 4, we will describe the mathematical formulation of each of the terms in Eq. (1), (2) and (5), and how SedTrace generates the corresponding code. We will use `SimpleFe` to illustrate this process.


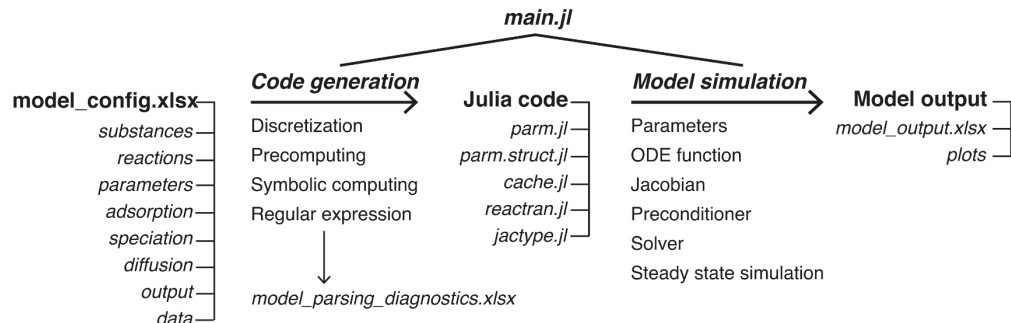

**Figure 1. SedTrace framework and workflow.**

**Table 1. The `substances` sheet of the `SimpleFe` model.**

| substance[1] | type | formula[2] | top_bc_type[3] | bot_bc_type[3] |
|---|---|---|---|---|
| POC | solid | CH2O | Robin | Neumann |
| FeOOH | solid | | Robin | Neumann |
| FeS | solid | | Robin | Neumann |
| SO4 | dissolved | | Robin | Neumann |
| TFe | dissolved_speciation | | Dirichlet | Neumann |
| H | dissolved_pH | | Dirichlet | Neumann |
| TCO2 | dissolved_pH | | Dirichlet | Neumann |
| TH2S | dissolved_pH | | Dirichlet | Neumann |

[1]Code name for the model substance. [2]Chemical formula that can be used to write chemically balanced equations in the `reactions` sheet.

[3]For model substances of type `dissolved_speciation`, the boundary conditions listed here are for the total dissolved concentration.

**Table 2. The `reactions` sheet of the `SimpleFe` model.**




| check[1] | label | equation[2] | rate[3] | Omega[3] |
|---|---|---|---|---|
| 1 | RFeOOHPOC | CH2O + 4*FeOOH + 8*H{+} = CO2 + 4*Fe{2+} + 7*H2O | FeOOH/(KFeOOH+FeOOH)* k_POC*POC | |
| 1 | RSO4POC | CH2O + 1/2*SO4{2-} + H{+} = CO2 + 1/2*H2S + H2O | SO4/(KSO4+SO4)*KFeOOH/ (KFeOOH+FeOOH)*k_POC*POC | |
| 1 | RFeOOHH2S | 2*FeOOH + H2S +4*H{+} = 2*Fe{2+} + S + 4*H2O | kFeOOHH2S*FeOOH*TH2S | |
| 1 | RFeS_pre | Fe{2+} + HS{-} = FeS+ H{+} | kFeSpre*(Omega_RFeS_pre-1) | Fe_aq*HS/(H*KspFeS) |

[1]Chemical balance is checked if `check` is set to 1. [2]Equations can be written using the code name or the chemical formula of model substances, or the code names and formulae of their dissolved and adsorbed species. [3]Rate and Omega expressions can be written only using the code name of the model substances or the code names of their dissolved and adsorbed species.

**Table 3. The `speciation` sheet of the `SimpleFe` model.**

| substance | dissolved[1] | formula[2] | equation[3] | logK |
|---|---|---|---|---|
| TFe | Fe_eq | Fe{2+} | Fe{2+} = Fe{2+} | 0.00 |
| TFe | FeCl_eq | FeCl{+} | Fe{2+} + Cl{-} = FeCl{+} | -0.12 |
| TFe | FeSO4_eq | (FeSO4)[1] | Fe{2+} + SO4{2-} = FeSO4 | 0.96 |
| TFe | FeCO3_eq | (FeCO3)[1] | Fe{2+} + CO3{2-} = FeCO3 | 3.65 |
| TFe | FeHS_eq | FeHS{+} | Fe{2+} + HS{-} = FeHS{+} | 5.40 |

[1]Code name of the dissolved species; must be different from the names of model substances in the `substances` sheet. [2]Chemical formula that can be used to write the chemical equations; must be different from the formulae of model substances in the `substances` sheet. [3]Equations should be written using formula, not code name.

**Table 4. The `adsorption` sheet of the `SimpleFe` model.**

| substance | dissolved[1] | adsorbed[2] | surface[3] | expression[4] | top_bc_type[5] | bot_bc_type[5] |
|---|---|---|---|---|---|---|
| TFe | TFe_dis | Fe_ads | POC | KFe_ads*POC*TFe_dis | Robin | Neumann |

[1]Code name of dissolved species that appears in the `speciation` sheet, or the code name of the total dissolved concentration. [2]Code name of adsorbed species. [3]Surface to be adsorbed onto; can be either the code name of a solid substance from the `substances` sheet, or left empty. [4]Expression can be written only using the code names of the dissolved and adsorbed species listed in the same row; adsorption parameters need to be supplied to the `parameters` sheet. [5]The boundary conditions of the adsorbed species, which should be the same for all adsorbed species of the same substance.

**Table 5. The `diffusion` sheet of the `SimpleFe` model.**

| model_name[1] | SedTrace_name[2] |
|---|---|
| SO4 | SO4{2-} |
| TFe_dis | Fe{2+} |

[1]Code name of the dissolved model substance, or its total dissolved concentration. [2]The corresponding name listed in SedTrace's database of the diffusion coefficients.



**Table 6. The `parameters` sheet of the `SimpleFe` model. A template of this sheet can be generated using `generate_parameter_template()`.**

| class[1] | type[2] | parameter[3] | value[4] | unit[5] | comment[5] |
|---|---|---|---|---|---|
| global | const | depth | 500 | m | water depth |
| global | const | salinity | 35 | psu | bottom water salinity |
| global | const | temp | 5 | Celsius | bottom water temperature |
| global | const | ds_rho | 2.6 | g cm^-3 | dry sediment density |
| grid | const | L | 50 | cm | model sediment section thickness |
| grid | const | Ngrid | 200 | integer | number of model grid |
| grid | function | gridtran | x | cm | grid transformation function |
| porosity | function | phi | 0.8 | dimensionless | porosity as a function of depth |
| porosity | const | phi_Inf | 0.7884 | dimensionless | porosity at infinite depth |
| burial | const | Fsed | 0.073 | g cm^-2 yr^-1 | total sediment flux |
| bioturbation | function | Dbt | 10*exp(-x/3) | cm^2/yr | bioturbation coefficient |
| bioirrigation | function | Dbir | 10*exp(-x/2) | yr^-1 | bioirrigation coefficient |
| speciation | const | KFe_ads | 1 | dimensionless | adsorption constant |
| speciation | const | Cl | 0.565772678 | mmol cm^-3 | Seawater Cl concentration |
| BoundaryCondition | const | delta | 5.00E-02 | cm | diffusive boundary layer |
| BoundaryCondition | const | FPOC0 | 0.31 | mmol cm^-2 yr^-1 | Flux of POC at the TOP |
| BoundaryCondition | const | FFeOOH0 | 0.023 | mmol cm^-2 yr^-1 | Flux of FeOOH at the TOP |
| BoundaryCondition | const | FFeS0 | 2.22045E-16 | mmol cm^-2 yr^-1 | Flux of FeS at the TOP |
| BoundaryCondition | const | SO4BW | 0.028 | mmol cm^-3 | Bottom water SO4 |
| BoundaryCondition | const | TFe_dis0 | 2.95E-08 | mmol cm^-3 | Concentration of TFe_dis at the TOP |
| BoundaryCondition | const | FTFe_ads0 | 0 | mmol cm^-2 yr^-1 | Flux of TFe_ads at the TOP |
| BoundaryCondition | const | pH0 | 7.59 | free pH scale | pH at the TOP |
| BoundaryCondition | const | TCO20 | 0.002345 | mmol cm^-3 | Concentration of TCO2 at the TOP |
| BoundaryCondition | const | TH2S0 | 2.22045E-16 | mmol cm^-3 | Concentration of TH2S at the TOP |
| Reaction | const | KspFeS | 10^(-3.2) | (mmol cm^-3)^-1 | apparent solubility of FeS |
| Reaction | const | KFeOOH | 3 | mmol cm^-3 | Monod constant FeOOH |
| Reaction | const | k_POC | 0.01 | yr^-1 | POC remineralization rate constant |
| Reaction | const | KSO4 | 0.001 | mmol cm^-3 | Monod constant of SO4 |
| Reaction | const | kFeOOHH2S | 4000 | (mmol cm-3)^-1 yr^-1 | rate const of H2S oxidation by FeOOH |
| Reaction | const | kFeSpre | 0.2*ds_rho | mmol cm^-3 yr^-1 | FeS precipitation rate constant |

[1]Class must be specified using one of the key words shown here following the same order. [2]Type can be `const` or `function` of depth x. [3]Code name for parameters. [4]To enter the value for the parameter, either supply a numerical value, or a function of depth, or a string expression that compute the value using other parameters, in which case the parameters being depended on must appear earlier in the table. The values shown here for the `SimpleFe` model is only for illustration. [5]Optional.

**Table 7. The `output` sheet of the `SimpleFe` model.**



| name[1] | expression[2] | conversion_profile[3] | unit_profile |
|---|---|---|---|
| POC | | 12/ds_rho/10 | wt% |
| FeOOH | | 88.85174/ds_rho/10 | wt% |
| FeS | | 87.91/ds_rho/10 | wt% |
| Fe | | 1.00E+06 | uM |
| SO4 | | 1.00E+03 | mM |
| pH | -log10(H) | 1.00E+00 | free scale |
| TCO2 | | 1.00E+03 | mM |
| TH2S | | 1.00E+03 | mM |
| TA[4] | | 1.00E+03 | mM |

[1]Code name for output variables; can be the same as the model substances, or new variable names. [2]Expressions to compute new variables, if needed; the expression can use model parameters listed in `parm.jl`. [3]Multiplication factors that convert default SedTrace units to custom units in `unit_profile`; can use model parameters listed in `parm.jl`; must not be empty (use 1 instead). [4]TA is computed internally and do not need an `expression`.

## 3 Grid, transport and boundary conditions

### 3.1 Grid

To generate the model grid, the user specifies the number of grid (`Ngrid`), the length of the sediment domain (`L`), and a grid transformation function (`gridtran`) in the `parameters` sheet. SedTrace internally creates a uniform grid between 0 and `L` cm, and uses `gridtran` to transform it to the user desired grid (using function `x->x` will preserve the original uniform grid). SedTrace allows the user to supply parameters as constants or functions of depth $x$, as labeled in the `type` column of the `parameters` sheet (Table 6), for example, `gridtran` as a `function` in this case. SedTrace will convert the function string to Julia function, and use `broadcast()` to vectorize the function if necessary. The generated model grid code for `SimpleFe` using a `gridtran` that has small grid spacing closer to the SWI is:

```
#-----------------------------------------------
# grid parameters
#-----------------------------------------------
L = 50.0 # cm # model sediment section thickness
Ngrid = 200 # integer # number of model grid
ξ = range(0, step = L / (Ngrid), length = Ngrid + 1) # cm # uniform grid
xᵥ = broadcast(x -> L * (exp(x / 5) - 1) / (exp(L / 5) - 1), ξ) # cm # non-uniform grid
transformation
x = (xᵥ[2:(Ngrid+1)] .+ xᵥ[1:Ngrid]) / 2 # cm # cell center
dx = xᵥ[2:(Ngrid+1)] .- xᵥ[1:Ngrid] # cm # cell volume
```

### 3.2 Advection

The advective flux in Eq. (2) is:

$$F_i^{\xi,adv} = \phi^\xi v^\xi C_i^\xi, \tag{6}$$

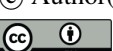



where $v^\xi$ is the phase velocity. To calculate $v^\xi$, SedTrace makes two assumptions (Boudreau, 1997; Meysman et al., 2003; Berner, 1980): sediment compaction is at steady state and therefore the volume fractions of fluid $\phi^f$ and solid $\phi^s (= 1 - \phi^f)$ are functions of depth but not time; the final burial velocities of fluid and solid are the same, $v^f_\infty = v^s_\infty$, without externally forced pore water advection. Using the user supplied porosity function $\phi^f$ (phif), porosity at burial depth $\phi^f_\infty$ (phif_Inf), density of dry sediments $\rho^s$ (ds_rho), and solid burial flux $F^s_{total}$ (Fsed) in parameters, SedTrace computes the phase velocities:

$$v^s_\infty = \frac{F^s_{total}}{\rho^s(1-\phi^f_\infty)},\tag{7}$$

$$v^f(x) = \frac{\phi^f_\infty v^s_\infty}{\phi^f(x)},\tag{8}$$

$$v^s(x) = \frac{(1-\phi^f_\infty)v^s_\infty}{1-\phi^f(x)}.\tag{9}$$

In SimpleFe the resulting code is:

```
#------------------------------------------------
# porosity parameters
#------------------------------------------------
phi_Inf = 0.7884 # dimensionless # porosity at infinite depth
phif = broadcast(x -> 0.8 + (0.9 - 0.8) * exp(-x / 2), x) # dimensionless # fluid volume
fraction
phis = 1.0 .- phif # dimensionless # solid volume fraction
#------------------------------------------------
# phase velocity parameters
#------------------------------------------------
Fsed = 0.073 # g cm^-2 yr^-1 # total sediment flux
w_Inf = Fsed / ds_rho / (1 - phi_Inf) # cm yr^-1 # solid sediment burial velocity at
infinite depth
uf = phi_Inf * w_Inf ./ phif # cm yr^-1 # pore water burial velocity
us = Fsed / ds_rho ./ phis # cm yr^-1 # solid sediment burial velocity
```

### 3.3 Diffusion

For solid substances SedTrace considers the diffusive flux due to bioturbation (Boudreau, 1997):

$$F^{s,diff}_i = -\phi^s D^b \frac{\partial C^s_i}{\partial x},\tag{10}$$

where $D^b$ (Ds) is the bioturbation coefficient as a function of depth specified in parameters:

```
#------------------------------------------------
# bioturbation parameters
#------------------------------------------------
Ds = broadcast(x -> 10 * exp(-x / 3), x) # cm^2 yr^-1 # Bioturbation coefficient
```

For dissolved substances SedTrace considers molecular diffusion $D^{md}_i$ corrected for the tortuosity factor $\theta^2$ (Boudreau, 1997):

$$F^{f,diff}_i = -\phi^f \frac{D^{md}_i}{\theta^2}\frac{\partial C^f_i}{\partial x},\tag{11}$$

$$\theta^2 = \frac{1}{1-\ln(\phi^f(x)^2)},\tag{12}$$





Boudreau (1997) parameterized $D_i^{md}$ at infinite dilution and atmospheric pressure ($P_{atm}$) as linear functions  ($m_0 + m_1 T$) of temperature ($T$ in Celsius) for selected dissolved substances. In this case SedTrace computes $D_i^{md}$ at user specified *in situ T*, salinity ($S$) and pressure ($P$) using the Stokes-Einstein relationship (Li and Gregory, 1974):

$$D_i^{md} = (m_0 + m_1 T) \frac{\mu(0,T,P_{atm})}{\mu(S,T,P)},$$  (13)

where $\mu$ is the dynamic viscosity of pore water as a function of $T$, $S$ and $P$. However, if only the diffusion coefficient $D_i^{md^{25^oC}}$

at infinite dilution, 25°C and atmospheric pressure is known, SedTrace computes

$$D_i^{md} = D_i^{md^{25^oC}} \frac{\mu(0,25,P_{atm})}{\mu(S,T,P)} \frac{T+273.15}{298.15}.$$  (14)

The values of $m_0$, $m_1$ and $D_i^{md^{25^oC}}$ of selected dissolved substances compiled by Boudreau (1997) are stored in the database file `/src/diffusion.xlsx`. If a model substance is in the database, then the user simply needs to list it in the `diffusion` sheet, where their `model_name` given by the user needs to match the `SedTrace_name` in the database (Table

5). SedTrace will compute $D_i^{md}$ automatically. If the model substance is not in the database, the user can either modify the database file to include it, or directly supply $D_i^{md}$ to `parameters`.

The example code for diffusion coefficient is:

```
#----------------------------------------------
# solute diffusivity
#----------------------------------------------
DSO4 = 1.8034511184582917E+02 ./ (1.0 .- 2log.(phif)) # cm^2 yr^-1 # Sediment diffusion
coefficient
```

### 3.4 Total physical transport

SedTrace collects the discretized advective and diffusive fluxes as an interior transport matrix **A** (`Am`) by calling the

`fvcf()` function, which performs the FV-CF discretization. The **AC** term in Eq. (5) is then computed. The code for the transport of `SO4` (`AmSO4`) is:

```
#----------------------------------------------
# Interior transport matrix
#----------------------------------------------
AmSO4 = fvcf(phif, DSO4, uf, dx, Ngrid) #  # Interior transport matrix of SO4
#----------------------------------------------
# Transport term A*C
#----------------------------------------------
mul!(dSO4, AmSO4, SO4) # dSO4 is the rate of change
```

### 3.5 Bio-irrigation

Bio-irrigation sources/sinks of dissolved substance is modeled as a non-local exchange (Boudreau, 1997):

$$T_i^{bio} = \alpha(C_i^f - C_i^{BW}),$$  (15)

where $\alpha$ (`alpha`) is the bio-irrigation coefficient as a function of depth $x$ and $C_i^{BW}$ is the bottom water concentration, both supplied to `parameters`. Previous studies have suggested that the bio-irrigation coefficient $\alpha$ is also substance-specific





(Meile et al., 2005), but it is unknown how this effect should be parameterized. SedTrace thus does not consider it at the moment.

The code for the biological transport term of SO4 is:

```
#------------------------------------------
# bioirrigation parameters
#------------------------------------------
alpha = broadcast(x -> 10 * exp(-x / 2), x) # yr^-1 # Bioirrigation coefficient
#------------------------------------------
# biological transport
#------------------------------------------
@.. dSO4 += alpha * (SO4BW - SO4)
```

**3.6 Boundary conditions**

At the SWI, for a solid substance ($C_i^s$) the users can specified a Robin boundary condition using the incoming flux ($FC_i^{s0}$) (Boudreau, 1997):

$$-\phi^s D^b \frac{\partial C_i^s}{\partial x}\Big|_{x=0} + \phi^s v^s\, C_i^s\big|_{x=0} = FC_i^{s0}, \tag{16}$$

or a Dirichlet boundary condition using the concentration at the SWI ($C_i^{s0}$):

$$C_i^s\big|_{x=0} = C_i^{s0}, \tag{17}$$

At the SWI, for a dissolved substance ($C_i^f$) the users can specified a Robin boundary condition assuming the existence of a Diffusive Boundary Layer (DBL) of thickness $\delta$ and the overlying bottom water concentration of $C_i^{BW}$ (Boudreau, 1997):

$$-\phi^f \frac{D_j^{i,md}}{\theta^2} \frac{\partial C_i^f}{\partial x}\Big|_{x=0} + \phi^f v^f\, C_i^f\big|_{x=0} = \frac{D_j^{i,md}}{\delta}(C_i^{BW} - C_i^f\big|_{x=0}), \tag{18}$$

or a Dirichlet boundary condition using the concentration at the SWI ($C_i^{f0}$):

$$C_i^f\big|_{x=0} = C_i^{f0}. \tag{19}$$

At the bottom of the model domain ($x = L$), the users can specify either a Neumann or a Dirichlet boundary condition for model substances:

$$\frac{\partial C_i^\xi}{\partial x}\Big|_{x=L} = 0, \tag{20}$$

$$C_i^\xi\big|_{x=L} = C_i^{\xi L}. \tag{21}$$

Here Eq. (20) assumes no concentration gradient at $x = L$.

The user specifies the type of boundary conditions in `substances`, and provides the necessary parameters in `parameters` (Table 6). SedTrace will format the boundary conditions as:

$$\alpha_1 C_i^\xi + a_2 \frac{\partial C_i^\xi}{\partial x} = a_3\,. \tag{22}$$





For example, setting the Robin boundary condition Eq. (18) at the SWI for `SO4` (Table 1), SedTrace will compute $\beta = \frac{D_j^{i,md}}{\delta}$ (`beta`), which is the mass transfer velocity, and $\alpha_1^0 = \beta + \phi^f v^f$ , $a_2^0 = -\phi^f \frac{D_j^{i,md}}{\theta^2}$ and $a_3^0 = \beta C_i^{BW}$ using $\delta$ (`delta`) and $C_i^{BW}$ (`SO4BW`) from the `parameters`. The Neumann bottom boundary condition is simply $\alpha_1^L = 0$, $a_2^L = 1$ and $a_3^L = 0$. SedTrace collects these coefficients into a `Tuple` $((a_1^0, a_2^0, a_3^0),\;\; (a_1^L, a_2^L, a_3^L))$, which is passed to `fvcf_bc()` to generate the homogeneous (`BcAM`, **B**) and non-homogenous (`BcCm`, **b**) parts of the boundary condition based on the FV-CF

discretization. SedTrace then updates the right-hand side of Eq. (5) by adding the $\mathbf{BC} + \mathbf{b}$ term. The code for `SO4` is:

```
#---------------------------------------------
# assemble boundary conditions
#---------------------------------------------
BcSO4 = (
    (betaSO4 + phif[1]uf[1], -phif[1]DSO4[1], betaSO4 * SO4BW),
    (0.0, 1.0, 0.0),
) #  # Boundary condition of SO4
#---------------------------------------------
# Boundary transport matrix
#---------------------------------------------
BcAmSO4, BcCmSO4 = fvcf_bc(phif, DSO4, uf, dx, BcSO4, Ngrid) #  # Boundary transport matrix
of SO4
#---------------------------------------------
# Boundary terms
#---------------------------------------------
dSO4[1] += BcAmSO4[1] * SO4[1] + BcCmSO4[1]
dSO4[Ngrid] += BcAmSO4[2] * SO4[Ngrid] + BcCmSO4[2]
```

## 4 Biogeochemical reactions

       Biogeochemical reactions in diagenetic models can be classified as kinetic or equilibrium reactions (Boudreau, 1997;

Meysman et al., 2003). The difference mainly lies in the reaction timescale: reactions that happen on much shorter timescales than physical transport are often treated as equilibrium rather than kinetic reactions. The user is free to add the kinetic reactions, while SedTrace has specific user interface for two types of equilibrium reactions: acid dissociation for pH modeling, and aqueous complexation-sorption for speciation modeling. SedTrace uses the Direct Substitution Approach (DSA) to handle the equilibrium reactions, which performs better than other approaches when dealing with highly coupled and stiff biogeochemical

reaction network (Meysman et al., 2003).

### 4.1 Kinetic reactions

       The summed rate of kinetic reactions for substance $C_i^\xi$ is:

$$R_i = \sum_\eta \phi^\eta/\phi^\xi \sum_k \nu_i^{\eta,k} r^{\eta,k}, \tag{23}$$

where $r^{\eta,k}$ is the rate of the $k$th reaction in unit of per volume of phase $\eta$, and $\nu_i^{\eta,k}$ is the stoichiometry of the $i$th substance in

this reaction. The unit of $R_i$ is in per volume of phase $\xi$. The convention of SedTrace is that homogeneous reactions between





dissolved substances are in unit of per volume of fluid, while heterogeneous reactions between dissolved and solid substances and homogenous reactions between the solid substances are in unit of per volume of solid. Conversion between fluid and solid concentration units are done using the conversion factor $\phi^f/\phi^s$ (fluid to solid, `pwtods`) or $\phi^s/\phi^f$ (solid to fluid, `dstopw`).

Kinetic reactions are added to the `reactions` sheet, including their chemical equations and rate expressions. 340 SedTrace parses the reactions and re-assembles them into Julia code, using Julia's Perl-compatible regular expression engine and the symbolic computing utility from the SymPy.jl package (Meurer et al., 2017).

### 4.1.1 Chemical equations

In SedTrace each model substance and their dissolved and adsorbed species need to have a code name and a chemical formula, supplied to the Excel sheets as noted in the Tables. The code names are used in the code, and we suggest only using 345 the Latin alphabet and "_" in the name. The formulae are necessary only when writing chemically balanced reaction equations. The code name and the chemical formula can be the same, if and only if the chemical formula is an allowed Julia variable name, such as `FeOOH` but not `Fe{2+}`. The chemical formula of a substance should be written in the form of `(E)[e](F)[f](G)[g]{h+}`, where `E/F/G` are compounds, `e/f/g` are their stoichiometric coefficients, and `h` is the number of charge. For example, the user can name organic matter `POM` and write its chemical formula as 350 `(CH2O)(NH3)[rNC](H3PO4)[rPC]`, where `rNC` and `rPC` are the N/C and P/C ratios. SedTrace allows parameterized stoichiometry, and the parameters `rNC` and `rPC` should be supplied to `parameters`.

SedTrace requires the user to write chemical equations in the form of `a*A + b*B = c*C + d*D`, where `a/b/c/d` are the stoichiometry coefficients of substances `A/B/C/D`, and the reactants are on the left hand side. The user has two options when writing this equations, decided by the `check` column in the `reactions` sheet (Table 2), which informs SedTrace if 355 the chemical balance of the equation should be checked. The user can write a chemically balanced equation using the chemical formulae of the model substances `A/B/C/D`, and set `check` to 1. Or the user can leave `check` empty and write a heuristic equation without considering chemical balance, using code names only or a mixture of code names and chemical formulae. SedTrace offers the second choice because it is common in diagenetic literature to write heuristic reactions. This happens because of the complexity of biogeochemical reactions in marine sediment, such that it is often not possible to know the exact 360 chemical formula or reaction mechanism. An example heuristic chemical equations is `POM = Carbon + rNC*NH3 + rPC*PO4` for organic matter remineralization.

SedTrace uses regular expressions to split the chemical equation into reactants and products, and identify the stoichiometry coefficients $\nu_i^{\eta,k}$ and charges of the model substances. SedTrace further parses the equation down to the level of individual elements, identifying the stoichiometry of each element. Here we show the parsing result of a more complex 365 version of the Fe reduction reaction in Table 2 `(CH2O)(NH3)[rNC](H3PO4)[rPC] + 4*FeOOH + (8+rNC-rPC)*H{+} = CO2 + rNC*NH4{+} + rPC*H2PO4{-} + 4*Fe{2+} + 7*H2O`:
`#------------------------------------------`



```
# Parsing the chemical equation into reactants/products
#------------------------------------------------
Row │ species                          stoichiometry  charge   role
    │ String                           String         String   String
────┼──────────────────────────────────────────────────────────────────
  │ (CH2O)(NH3)[rNC](H3PO4)[rPC]     -1             0        reactant
  │ FeOOH                            -4             0        reactant
  │ H                                -(8+rNC-rPC)   +1       reactant
  │ CO2                              1              0        product
  │ NH4                              rNC            +1       product
  │ H2PO4                            rPC            -1       product
  │ Fe                               4              +2       product
  │ H2O                              7              0        product
#------------------------------------------------
# Parsing the chemical equation into elements
# negative value indicates reactant
#------------------------------------------------
Row │ element  coef
    │ String   String
────┼──────────────────────────────
  │ C        -1
  │ H        -2
  │ O        -1
  │ N        (-rNC)
  │ H        (-3*rNC)
  │ H        (-3*rPC)
  │ P        (-rPC)
  │ O        (-4*rPC)
  │ Fe       -4
 │ O        -4
 │ O        -4
 │ H        -4
 │ H        (-rNC + rPC - 8)
 │ C        1
 │ O        2
 │ N        (rNC)
 │ H        (4*rNC)
 │ H        (2*rPC)
 │ P        (rPC)
 │ O        (4*rPC)
 │ Fe       4
 │ H        14
 │ O        7
```

Since the stoichiometry may be parameterized, SedTrace uses symbolic computing to check the chemical balance if check = 1. Parameters like rNC and rPC are converted to Julia symbols of type SymPy.Sym. The sums of charge and mass are computed symbolically, and errors are thrown if the sums are not zero. If check is empty SedTrace will parse the equation and identify the stoichiometric coefficients, but will not check the chemical balance.

### 4.1.2 Reaction rates

The string expressions of reaction rates, $r^{\eta,k}$ in Eq. (23), supplied to the reactions sheet are copy-pasted as Julia code directly, and therefore they should be written only using the code names. For dissolution and precipitation reactions, the





user can add the definition of saturation state in a separate column `Omega`. SedTrace will allow dissolution or precipitation only when `Omega<1` or `Omega>1` respectively. However, an `if Omega>1 then precipitate` statement, *i.e.*, a

Heaviside step function, in the code induces numerical discontinuity, which can hurt numerical performance, especially when applying automatic differentiation to the code. SedTrace thus approximates the Heaviside step function using the Logistic function which is continuously differentiable:

$$r'_{pre} = (\tfrac{1}{2}\tanh(\tau \tfrac{\Omega-1}{2}) + \tfrac{1}{2})r_{pre}, \tag{24}$$

where $r_{pre}$ is a precipitation rate and the parameter $\tau$ controls how close the approximation is. By default $\tau = 10^3$ which

results in a sharp transition near saturation. For example, the code for `FeS` precipitation rate `RFeS_pre` is:

```
@.. Omega_RFeS_pre = Fe_eq * HS / (H * KspFeS) # saturation state for precipitation
@.. RFeS_pre =
    (tanh(1e3 * (Omega_RFeS_pre - 1.0)) / 2 + 0.5) *
    (kFeSpre * (Omega_RFeS_pre - 1)) # precipitation rate
```

To compute the summed reaction rate $R_i$ in Eq. (23), SedTrace collects the stoichiometry coefficients $\nu_i^{\eta,k}$ of the model substances in each kinetic reaction after parsing the chemical equations, and applies the appropriate unit conversion factors (`dstopw` or `pwtods`). For example, code for the summed reaction rate `S_FeOOH` of Fe oxide is:

```
@.. S_FeOOH = -4 * RFeOOHPOC + -2 * RFeOOHH2S
```

## 4.2 pH modeling

SedTrace models pH using the DSA outlined by Hofmann et al., (2008, 2009, 2010). The dynamic equation for proton concentration ($[H^+]$, free scale) is:

$$\tfrac{\partial}{\partial t}[H^+] = (\tfrac{\partial}{\partial t}TA - \sum_l \tfrac{\partial TA}{\partial EI_l}\tfrac{\partial}{\partial t}EI_l)/\tfrac{\partial TA}{\partial[H^+]}, \tag{25}$$

where $TA$ is the total alkalinity (TA), and $EI_l$ is the $l$th Equilibrium Invariant (EI). The EIs are composite variables, such as the total dissolved inorganic carbon (TCO$_2$). They are so named because they are invariant with respect to the equilibrium

reaction rates.

The full definition of TA in seawater (Dickson et al., 2007) is:

$$TA = [HCO_3^-] + 2[CO_3^{2-}] + [B(OH)_4^-] + [OH^-] + [HPO_4^{2-}] + 2[PO_4^{3-}] + [H_3SiO_4^-] + [NH_3] + [HS^-] - [H^+] -$$
$$[HF] - [HSO_4^-] - [H_3PO_4], \tag{26}$$

To include all these species, SedTrace provides the following EIs:

$$TCO_2 = [HCO_3^-] + [CO_3^{2-}] + [CO_2], \tag{27}$$

$$TH_2S = [H_2S] + [HS^-], \tag{28}$$

$$TH_3BO_3 = [B(OH)_3] + [B(OH)_4^-], \tag{29}$$

$$TNH_4 = [NH_3] + [NH_4^+], \tag{30}$$

$$TH_3PO_4 = [H_3PO_4] + [H_2PO_4^-] + [HPO_4^{2-}] + [PO_4^{3-}], \tag{31}$$

$$THSO_4 = [HSO_4^-] + [SO_4^{2-}], \tag{32}$$





$$THF = [HF] + [F^-] , \qquad (33)$$

$$TH_4SiO_4 = [H_4SiO_4] + [H_3SiO_4^-]. \qquad (34)$$

It is usually unnecessary to include the entire set in diagenetic models. The user is free to choose any subset of this collection. The user adds the selected EIs to the `substances` sheet and specify the `type` as `dissolved_pH`. Apart from

supplying the boundary conditions, all other aspect of pH modeling is handled internally by SedTrace requiring no user input. Based on the user's choice, SedTrace defines TA as a subset of the full definition in Eq. (26). For example, in `SimpleFe` TCO$_2$ and TH$_2$S are selected (Table 1), then $TA = [HCO_3^-] + 2[CO_3^{2-}] + [HS^-] + [OH^-] - [H^+]$. H$^+$ and OH$^-$ are always included by default.

SedTrace uses `EquilibriumInvariant` to store the information of the EIs, including the analytical expressions

to compute the concentrations of the individual species listed in Eq. (27) to (34), their coefficients in Eq. (26), and the expressions to compute $\frac{\partial TA}{\partial EI_l}$ and $\frac{\partial TA}{\partial [H^+]}$, for example TCO$_2$:

```
struct EquilibriumInvariant
    name::String # name of EI
    species::Vector{String} # species
    charge::Vector{String} # charges of the species
    expr::Vector{String} # expression to compute the species concentration
    coef::Vector{String} # coefficient of species in TA definition
    dTAdEI::String # expression to compute dTA/dEI
    dTAdH::String # expression to compute this EI's contribution to dTA/dH
    diss_const::Vector{String} # acid dissociation constants
end
# TCO2 for example
EquilibriumInvariant(
        "TCO2",
        ["HCO3", "CO3", "CO2"],
        ["{-}","{2-}",""],
        [
            "H * KCO2 * TCO2 / (H^2 + H * KCO2 + KCO2 * KHCO3)",
            "KCO2 * KHCO3 * TCO2 / (H^2 + H * KCO2 + KCO2 * KHCO3)",
            "H^2 * TCO2 / (H^2 + H * KCO2 + KCO2 * KHCO3)",
        ],
        ["1", "2", "0"],
        "KCO2*(H + 2*KHCO3)/(H^2 + H*KCO2 + KCO2*KHCO3)",
        "-KCO2*TCO2*(H^2 + 4*H*KHCO3 + KCO2*KHCO3)/(H^2+ H*KCO2 + KCO2*KHCO3)^2",
        ["KCO2","KHCO3"]
        )
```

SedTrace stores precomputed dissociation constants (on the free proton scale) on high resolution grids of salinity, temperature and pressure in `/src/dissociation_constant.jld2` following the *Guide to best practices for ocean CO$_2$ measurements* (Dickson et al., 2007) as implemented in the `seacarb` package (Gattuso et al., 2021). During code

generation, SedTrace will compute the dissociation constants at the *in situ* salinity, temperature and pressure by interpolation. In the case of `SimpleFe`, the dissociation constant of H$_2$O (`KH2O`), the first (`KCO2`) and second (`KHCO3`) dissociation constants of H$_2$CO$_3$ and the first dissociate constant of H$_2$S (`KH2S`) are computed:

```
#----------------------------------------------
# Acid dissociation constants
```





```
#----------------------------------------------
      KH2O = 7.9445878598109790E-15 #H 1th dissociation constant
      KCO2 = 8.3698755808176183E-07 #TCO2 1th dissociation constant
      KHCO3 = 4.6352156109843975E-10 #TCO2 2th dissociation constant
      KH2S = 1.2845618784784923E-07 #H2S 1th dissociation constant
```

Given the concentrations of EIs and proton, SedTrace computes the concentrations of the individual species, and their

transport and boundary condition terms following Sect. 3. SedTrace uses species-specific diffusion coefficients. Many

diagenetic models transport EIs and TA using the diffusion coefficients of the dominant species or a fixed weighted average

of the diffusion coefficients of the individual species. However, studies have shown that this may lead to modeled pore water

pH being different from when using species-specific diffusion coefficient (Luff et al., 2001; Jourabchi et al., 2005). In the

`SimpleFe` example, code for the transport and boundary conditions of $HCO_3^-$ is:

```
@.. HCO3 = H * KCO2 * TCO2 / (H^2 + H * KCO2 + KCO2 * KHCO3)
mul!(HCO3_tran, AmHCO3, HCO3)
HCO3_tran[1] += BcAmHCO3[1] * HCO3[1] + BcCmHCO3[1]
HCO3_tran[Ngrid] += BcAmHCO3[2] * HCO3[Ngrid] + BcCmHCO3[2]
@.. HCO3_tran += alpha * (HCO30 - HCO3)
```

SedTrace then computes the transport and kinetic reaction terms of TA and EIs by summing over the species:

$$\frac{\partial}{\partial t}\phi^f EI_l + \sum_m T_{EI_l^m} = \sum_\eta \phi^\eta \sum_k (\sum_m \nu_{EI_l^m}^{\eta,k}) r^{\eta,k}, \tag{35}$$

$$\frac{\partial}{\partial t}\phi^f TA + \sum_n \zeta_n T_{TA^n} = \sum_\eta \phi^\eta \sum_k (\sum_n \zeta_n \nu_{TA^n}^{\eta,k}) r^{\eta,k}, \tag{36}$$

where $EI_l^m$ is the $m$th species of $EI_l$, $TA^n$ is the $n$th species in the definition of $TA$ and $\zeta_n$ is its coefficient in Eq. (26), $\nu_{EI_l^m}^{\eta,k}$

and $\nu_{TA^n}^{\eta,k}$ are the stoichiometric coefficients of these species in the kinetic reaction $r^{\eta,k}$ respectively.

The user can use the individual species when writing the chemical equations of the kinetic reaction rates. SedTrace

will parse the equations and identify $\nu_{EI_l^m}^{\eta,k}$ and $\nu_{TA^n}^{\eta,k}$ automatically. For example, SedTrace recognizes that for the reaction

`RFeS_pre` in Table 2, the stoichiometric coefficient of $TH_2S$ is $\nu_{HS^-} = -1$, and the stoichiometric coefficient of TA is

$\nu_{HS^-} - \nu_{H^+} = -2$. For the `SimpleFe` model the reaction-transport code for TA and EIs is:

```
# Transport of EIs
      @.. dTCO2 = HCO3_tran + CO3_tran + CO2_tran
      @.. dTH2S = H2S_tran + HS_tran
      # Transport of TA
      @.. TA_tran = -1 * H_tran + 1 * OH_tran
@.. TA_tran += 1 * HCO3_tran + 2 * CO3_tran + 0 * CO2_tran
      @.. TA_tran += 0 * H2S_tran + 1 * HS_tran
      # Kinetic reaction rates of EIs
      @.. S_TCO2 = 1 * RFeOOHPOC * dstopw + 1 * RSO4POC * dstopw
      @.. S_TH2S =
1 / 2 * RSO4POC * dstopw + -1 * RFeOOHH2S * dstopw + -1 * RFeS_pre
      # Kinetic reaction rates of TA
      @.. S_TA =
         8 * RFeOOHPOC * dstopw + 1 * RSO4POC * dstopw + 4 * RFeOOHH2S * dstopw +
         -1 * RFeS_pre
@.. S_TA += -1 * RFeS_pre
```





SedTrace does not explicitly model TA, rather Eq. (35) and (36) are substituted back into Eq. (25) to eliminate the TA terms to arrive at a diagenetic equation of $[H^+]$:

$$T_H = (\sum_n \zeta_n T_{TA^n} - \sum_l \frac{\partial TA}{\partial EI_l} \sum_m T_{EI_l^m})/\frac{\partial TA}{\partial [H^+]}, \tag{37}$$

$$R_H = \sum_\eta \phi^\eta/\phi^f \sum_k (\sum_n \zeta_n \nu_{TA^n}^{\eta,k} - \sum_l \frac{\partial TA}{\partial EI_l} \sum_m \nu_{EI_l^m}^{\eta,k})/\frac{\partial TA}{\partial [H^+]} r^{\eta,k}, \tag{38}$$

$$\frac{\partial}{\partial t} \phi^f [H^+] + T_H = \phi^f R_H. \tag{39}$$

In the terminology of Hofmann et al., (2010), $-\frac{\partial TA}{\partial [H^+]}$ is the buffer factor and $\nu_H^{\eta,k} = -(\sum_n \zeta_n \nu_{TA^n}^{\eta,k} - \sum_l \frac{\partial TA}{\partial EI_l} \sum_m \nu_{EI_l^m}^{\eta,k})$ is the *fractional* stoichiometric coefficient of proton in the kinetic reaction $r^{\eta,k}$. Equation (37) to (39) shows that the advantage of DSA is that the rate of pH change can be clearly partitioned at the level of individual species and reactions. For the `SimpleFe` model the code for Eq. (37) to (39) is:

```
#   dTA/dEIs
      @.. dTA_dTCO2 = KCO2 * (H + 2 * KHCO3) / (H^2 + H * KCO2 + KCO2 * KHCO3)
      @.. dTA_dTH2S = KH2S / (H + KH2S)
      #   dTA/dH
      @.. dTA_dH = -(H^2 + KH2O) / H^2
@.. dTA_dH +=
          -KCO2 * TCO2 * (H^2 + 4 * H * KHCO3 + KCO2 * KHCO3) /
          (H^2 + H * KCO2 + KCO2 * KHCO3)^2
      @.. dTA_dH += -KH2S * TH2S / (H + KH2S)^2
      #   transport of proton
@.. dH = TA_tran
      @.. dH -= dTCO2 * dTA_dTCO2
      @.. dH -= dTH2S * dTA_dTH2S
      @.. dH = dH / dTA_dH
      #   kinetic reaction rates of proton
@.. S_H = S_TA
      @.. S_H -= S_TCO2 * dTA_dTCO2
      @.. S_H -= S_TH2S * dTA_dTH2S
      @.. S_H = S_H / dTA_dH
```

### 4.3 Speciation modeling

In the DSA of speciation modeling, SedTrace models the total concentration of a model substance ($M^T$ in unit of per volume fluid), which is the sum of the total dissolved ($M^f$ in unit of per volume fluid) and total adsorbed ($M^s$ in unit of per volume solid) concentrations:

$$M^T = M^f + \frac{\phi^s}{\phi^f} M^s, \tag{40}$$

and $M^f$ and $M^s$ are themselves sums of the concentrations of individual dissolved and adsorbed species respectively.

To indicate speciation modeling is required for a model substance, the user needs to specify `dissolved_speciation` in the `type` column in the `substances` sheet (Table 1), and provide the dissolved and adsorbed speciation information in the `speciation` (Table 3) and `adsorption` (Table 4) sheets. The name given in the `substances` sheet is the code name of the total concentration $M^T$. Internally SedTrace will set the code names of the total





dissolved $M^f$ and total adsorbed $M^s$ concentrations by appending the postfix `_dis` and `_ads` to the code name of the model
substance. For example, the user names the total Fe `TFe` in `SimpleFe`, and SedTrace will name the total dissolved and
adsorbed Fe `TFe_dis` and `TFe_ads` respectively. The code names (column `dissolved`) and chemical formulae (column
`formula`) of the individual dissolved species should be supplied to `speciation` (Table 3). The code names (column
`adsorbed`) of the individual adsorbed species should be supplied to `adsorption` (Table 4) but no chemical formula for
the adsorbed species is needed.

The user can add dissolved speciation reaction of a trace element $M$ of the following type to column `equation` in
`speciation` (Pierrot and Millero, 2017):

$$M + q \times L_p \Leftrightarrow M(L_p)_q, \tag{41}$$

which describes complexation with the $p$th dissolved ligand $L_p$ to form aqueous species $M(L_p)_q$ and $q$ is the number of the
ligand in the complex. Assuming local equilibrium, the concentration of the complexed species is:

$$[M(L_p)_q] = K_{M(L_p)_q}[M][L_p]^q, \tag{42}$$

where $K_{M(L_p)_q}$ is the *apparent* equilibrium constant supplied to column `logK`, and $[M]$ is the concentration of the *base*
species $M$. $[M^f]$ is the sum of the concentrations of the base and complexed species:

$$[M^f] = [M](1 + \sum_p \sum_q K_{M(L_p)_q}[L_p]^q), \tag{43}$$

In the `speciation` sheet, The base species is indicated by writing a trivial `equation` such as `Fe{2+}=Fe{2+}`
with `logK=0` in the `SimpleFe` model where free $Fe^{2+}$ is the base species of `TFe` (Table 3). SedTrace will parse and check
the chemical balance of the equations as discussed in Sect. 4.1.1. The dissolved ligands have to be modeled or specified by the
users. In `SimpleFe` $[HS^-]$ is computed as part of the pH model, and the concentration of $Cl^-$ is not modeled but deemed a
constant and supplied to `parameters`. SedTrace 1.0 does not model the speciation of the ligands, and thus $[L_p]$ in Eq. (42)
refers to the total dissolved ligand concentrations. Future SedTrace versions are expected to include ligand speciation.

The formulation of adsorption in the literature of diagenetic modeling is diverse (Boudreau, 1997; Berner, 1980;
Wang and Van Cappellen, 1996; Katsev et al., 2006). The challenge lies in the fact that the surface properties of sedimentary
particles are poorly known. SedTrace thus does not constrain the formulation of adsorption, and lets the user specify the
concentrations of the adsorbed species directly.

In the `adsorption` sheet (Table 4), each row should list the adsorption of one dissolved species onto one surface.
The user names the adsorbed species in the `adsorbed` column, the dissolved species to be adsorbed in the `dissolved`
column, and the surface to be adsorbed onto in the `surface` column. All three columns should contain code names only. The
user then supplies a mathematical expression $f^{ads}$ to compute the adsorbed species concentration in the `expression`
column, as a function of the concentrations of the dissolved species and surface:

$$[M^s \equiv S_\kappa^\lambda] = f^{ads}([M^{dis}], [\equiv S_\kappa]), \tag{44}$$





where $[M^s \equiv S_\kappa^\lambda]$ is the concentration of the $\lambda$th dissolved species adsorbed onto the $\kappa$th particle surface $\equiv S_\kappa$. The dissolved species $M^{dis}$ can be one of $M$, $M(L_p)_q$ or $M^f$.

The term surface is used heuristically here, and can refer to any modeled solid substance. For example, in the `SimpleFe` model (Table 4), Fe is adsorbed onto `POC`, and the `expression` for the adsorbed species is `Fe_ads = KFe_ads*POC*TFe_dis`, where the adsorption is assumed to be indifferent to the dissolved speciation and thus the total dissolved Fe concentration `TFe_dis` ($M^f$) is used. The adsorption constant `KFe_ads` here is an *apparent* constant, and needs to be provided to the `parameters` sheet. The concentration of adsorbed species could be sometimes independent of any surface, for example, in the classic linear isothermal used by many diagenetic models (Berner, 1980), such that `Fe_ads = KFe_ads*TFe_dis`. In this case the `surface` column should be left empty.

SedTrace will group the adsorbed species by `surface`, and compute the total concentration adsorbed onto the surface $\equiv S_\kappa$ ($[M_\kappa^s]$, "empty surface" is a special surface) and the total concentration adsorbed onto all surfaces ($[M^s]$):

$$[M_\kappa^s] = \sum_\lambda [M^s \equiv S_\kappa^\lambda], \tag{45}$$

$$[M^s] = \sum_\kappa [M_\kappa^s], \tag{46}$$

Internally SedTrace creates a code name for $M_\kappa^s$ by appending the postfix `ads_surface` to the substance name, for example, `TFe_ads_POC` for the total Fe adsorbed onto POC. If the surface is empty, the code name for this adsorbed species is created by appending the postfix `_ads_nsf` to the substance name, for example `TFe_ads_nsf` for the linear isothermal case.

To generate the speciation code, SedTrace solves Eq. (40) to (46) using the `solveset` function of solving systems of symbolic nonlinear equations from SymPy.jl (Meurer et al., 2017). To do so the user supplied expression $f^{ads}$ needs to be analytically invertible. The results is a set of symbolic expressions to compute $[M]$, $[M(L_p)_q]$, $[M^f]$, $[M^s \equiv S_\kappa^\lambda]$, $[M_\kappa^s]$ and $[M^s]$, using $[M^T]$, $[L_p]$, $[\equiv S_\kappa]$ and the equilibrium constants, which are converted to Julia code.

The symbolic derivation starts by rewriting Eq. (44) to substitute $[M]$ and $[M(L_p)_q]$ by $[M^f]$ using Eq. (42) and (43). Together with Eq. (40), (45) and (46), SedTrace derives an expression to compute $[M^f]$ using $[M^T]$ and $[\equiv S_\kappa]$. In the case of `TFe_dis` in the `SimpleFe` model the code is:

```
# Concentrations of total dissolved species
@.. TFe_dis = TFe / (KFe_ads * POC * dstopw + 1)
```

Substituting this expression back to Eq. (42) and (43), SedTrace derives the expressions to compute $[M]$ and $[M(L_p)_n]$ using $[M^f]$ and $[L_p]$. The code for the individual dissolved Fe species is:

```
# Concentrations of the individual dissolved species
@.. Fe_aq =
    3.98107170553497e-6 * TFe_dis / (
        0.01778279410038921 * CO3 + 3.019951720402014e-6 * Cl + 1.0 * HS +
        3.630780547701011e-5 * SO4 + 3.98107170553497e-6
    )
@.. FeCl_aq =
```



```
3.019951720402014e-6 * Cl * TFe_dis / (
             0.01778279410038921 * CO3 + 3.019951720402014e-6 * Cl + 1.0 * HS +
             3.630780547701011e-5 * SO4 + 3.98107170553497e-6
         )
     @.. FeSO4_aq =
3.630780547701011e-5 * SO4 * TFe_dis / (
             0.01778279410038921 * CO3 + 3.019951720402014e-6 * Cl + 1.0 * HS +
             3.630780547701011e-5 * SO4 + 3.98107170553497e-6
         )
     @.. FeCO3_aq =
0.01778279410038921 * CO3 * TFe_dis / (
             0.01778279410038921 * CO3 + 3.019951720402014e-6 * Cl + 1.0 * HS +
             3.630780547701011e-5 * SO4 + 3.98107170553497e-6
         )
     @.. FeHS_aq =
1.0 * HS * TFe_dis / (
             0.01778279410038921 * CO3 + 3.019951720402014e-6 * Cl + 1.0 * HS +
             3.630780547701011e-5 * SO4 + 3.98107170553497e-6
         )
```

SedTrace then computes $[M^s \equiv S_\kappa^\lambda]$, $[M_\kappa^s]$ and $[M^s]$, using $[M]$, $[M^f]$, $[M(L_p)_n]$ and $[\equiv S_\kappa]$ based on Eq. (44) to (46), for example the code for the individual and total adsorbed Fe species is:

```
     # Concentrations of the individual adsorbed species
     @.. Fe_ads = KFe_ads * POC * TFe_dis
     # Concentrations of the total adsorbed species onto each surface
     @.. TFe_ads_POC = Fe_ads
# Concentrations of the total adsorbed species
     @.. TFe_ads = Fe_ads
```

In the next step, SedTrace computes the transport and reaction terms for $M^T$ using $M^f$ and $M^s$:

$$\frac{\partial}{\partial t} \phi^f [M^T] + T_{M^f} + T_{M^s} = \sum_\eta \phi^\eta \sum_k \nu_{M^T}^{\eta,k} r^{\eta,k}. \tag{47}$$

$T_{M^f}$ is the transport of total dissolved species summed over individual species. SedTrace uses the same diffusion coefficients for the dissolved species, as the diffusivities of the aqueous complexes are generally not known. $T_{M^s}$ is the transport of the total adsorbed species summed over individual species, which are subject to the same transport mechanism like normal solid substances. The user needs to supply two sets of boundary conditions when modeling $M^T$. The boundary conditions of $M^f$ is supplied to the substances sheet, and the boundary conditions of $M^s$ are supplied to the adsorption sheet.

When writing the chemical equations and rate expressions of the kinetic reactions, the user can use the code name or chemical formula of any of the following: $M$, $M(L_p)_q$, $M^f$, $M^s \equiv S_\kappa^\lambda$, $M_\kappa^s$, $M^s$ or $M^T$. SedTrace will parse the reactions and set the stoichiometric coefficient for $M^T$ ($\nu_{M^T}^{\eta,k}$). The reactive-transport code for TFe in the SimpleFe model is:

```
     # Transport and boundary condition of total dissolved concentration
     mul!(TFe_dis_tran, AmTFe_dis, TFe_dis)
TFe_dis_tran[1] += BcAmTFe_dis[1] * TFe_dis[1] + BcCmTFe_dis[1]
     TFe_dis_tran[Ngrid] += BcAmTFe_dis[2] * TFe_dis[Ngrid] + BcCmTFe_dis[2]
     # Transport and boundary condition of total adsorbed concentration
     mul!(TFe_ads_tran, AmTFe_ads, TFe_ads)
     TFe_ads_tran[1] += BcAmTFe_ads[1] * TFe_ads[1] + BcCmTFe_ads[1]
```





```
TFe_ads_tran[Ngrid] += BcAmTFe_ads[2] * TFe_ads[Ngrid] + BcCmTFe_ads[2]
      #  Transport of total concentration
      @.. dTFe = TFe_dis_tran * 1 + TFe_ads_tran * dstopw
      @.. dTFe += alpha * (TFe_dis0 - TFe_dis)
      #  Reaction rate of total concentration
@.. S_TFe = 4 * RFeOOHPOC * dstopw + 2 * RFeOOHH2S * dstopw + -1 * RFeS_pre
```

**4.4 Sediment age**

The user may sometimes need to know the sediment age. Use cases may rise for example when using the continuum reactivity model for organic carbon remineralization (Boudreau et al., 2008):

$$k_{POC} = \frac{\zeta}{a_0 + a_t}, \tag{48}$$

where $k_{POC}$ is the reaction rate constant of organic carbon, $\zeta$ is the parameter controlling the shape of the gamma distribution of reactivity, $a_0$ is the initial age of organic carbon, and $a_t$ is the duration of remineralization. The users may use sediment age in placement of $a_t$ (Freitas et al., 2021).

The diagenetic equation for sediment age is (Meile and Van Cappellen, 2005):

$$\frac{\partial}{\partial t}\phi^s Age + \frac{\partial}{\partial x}\phi^s v^s Age - \frac{\partial}{\partial x}\phi^s D^b \frac{\partial Age}{\partial x} = \phi^s. \tag{49}$$

The top boundary condition of sediment age can be specified as Robin or Dirichlet type. If specifying a zero incoming flux, the modeled age can be interpreted with respect to the incoming sediments the age of which is zero. If specifying a zero age at the SWI, the modeled age is set to zero whenever sediments make contact with the SWI. The bottom boundary condition of sediment age is specified as $\frac{\partial Age}{\partial x}\big|_{x=L} = \frac{1}{v^s|_{x=L}}$ to be consistent with the burial velocity. The use can add `Age` to the `substances` sheet with proper boundary conditions to enable sediment age modeling.

**5 Julia code files**

The code generated by SedTrace is assembled into 5 Julia files (Fig. 1): `parm.jl` and `parm.struct.jl` containing the model parameters, `cache.jl` containing the model cache, `reactran.jl` containing the reactive-transport code, and `jactype.jl` containing the sparsity pattern of the Jacobian matrix.

**5.1 Parameters**

The user supplied parameters in the `parameters` sheet, and those computed internally by SedTrace, are included in `parm.jl`. Parameters that are *directly* needed by the ODE function in Eq. (5) are further packed into a container `ParamStruct` within the `module Param` inside `parm.struct.jl` using Julia package Parameters.jl (Werder, 2022a). For instance, the diffusion coefficients are not needed by Eq. (5) directly and are thus only included in `parm.jl`. The transport matrix **A**, which incorporates the diffusion coefficients, is needed directly by Eq. (5) and thus included in `ParamStruct`.

For the `SimpleFe` model the code of the `ParamStruct` is:



```
@with_kw mutable struct ParamStruct{T} # T is a Parametric Type
# only showing a few entries
        TFeID::StepRange{Int64,Int64} = TFeID # index of TFe
        AmTFe_dis::Tridiagonal{T,Vector{T}} = AmTFe_dis # transport matrix of TFe
        TFe_dis0::T = TFe_dis0 # top boundary condition of TFe
        kFeSpre::T = kFeSpre # FeS precipitation rate constant
end
```

### 5.2 Cache

The code generation process creates many intermediate variables, which could cause repeated memory allocation during model simulation. Technically, they can be eliminated by substitution, but it would render the code unreadable. SedTrace preserves them for code clarity, and pre-allocates memories for them when initiating the model so the memories can be reused.

During code generation, SedTrace keeps track of which variables are intermediate variables. SedTrace adds them to a container `Reactran` within `module  Cache` inside `cache.jl`, and pre-allocates their memories using the package PreallocationTools.jl (Rackauckas and Nie, 2017). The cache is compatible with the `Dual` number type used by the package ForwardDiff.jl so automatic differentiation can be applied to the code (Revels et al., 2016).

```
mutable struct Reactran{T} # T is a Parametric Type
    TFe_dis::PreallocationTools.DiffCache{Array{T,1},Array{T,1}}
    FeCl_aq::PreallocationTools.DiffCache{Array{T,1},Array{T,1}}
     ...... # other intermediate variables
end
```

### 5.3 ODE function

The `reactran.jl` file contains a function `function(f::Cache.Reactran)(dC,  C,  parms::
Param.ParamStruct,  t)`, which includes the reactive-transport code to update $\frac{d}{dt}\mathbf{C}$ (dC) in Eq. (5) at time `t` in-place, given the current model state vector `C` and parameters `parms`. This function is compatible with the ODE solvers from DifferentialEquations.jl (Rackauckas and Nie, 2017) and the automatic differentiation tools in ForwardDiff.jl (Revels et al., 2016).

This function is assembled in the following sequence: (1) Unpack the parameters contained in `parms` using the package UnPack.jl (Werder, 2022b); (2) Retrieve pre-allocated cache; (3) Compute the transport and boundary conditions terms of model substances that do not require speciation; (4) Compute pH, EI speciation and related transport and boundary conditions terms; (5) Compute the speciation of model substances that require speciation, and their transport and boundary conditions terms; (6) Compute the individual kinetic reaction rates and the summed rates for model substances. A code skeleton for `SimpleFe` model is shown here:

```
function (f::Cache.Reactran)(dC, C, parms::Param.ParamStruct, t)
    #----------------------------------------------------------------
    #  Parameters
    #----------------------------------------------------------------
    @unpack TFeID,
```





```
      #...... other parameters
kFeSpre = parms
      #--------------------------------------------------------------
      #  Cache
      #--------------------------------------------------------------
      TFe_dis = PreallocationTools.get_tmp(f.TFe_dis, C)
#...... other caches
      #--------------------------------------------------------------
      #  Model state
      #--------------------------------------------------------------
      TFe = @view C[TFeID]
dTFe = @view dC[TFeID]
      #...... other model substances
      #--------------------------------------------------------------
      #  Transport of solid and dissolved substances See Section 3.4
      #--------------------------------------------------------------
#--------------------------------------------------------------
      #  pH code See Section 4.2
      #--------------------------------------------------------------
      #--------------------------------------------------------------
      #  Speciation code See Section 4.3
#--------------------------------------------------------------
      #  Concentrations of adsorbed/dissolved species
      #  Transport of adsorbed/dissolved species
      @.. dTFe = TFe_dis_tran * 1 + TFe_ads_tran * dstopw
      #--------------------------------------------------------------
#  Reaction code See Section 4.1
      #--------------------------------------------------------------
      # Individual reaction rates
      # Summed rates for model substances
      @.. S_TFe = 4 * RFeOOHPOC * dstopw + 2 * RFeOOHH2S * dstopw + -1 * RFeS_pre
# sum transport and reaction
      @.. dTFe += S_TFe
      return nothing
  end
```

### 5.4 Jacobian pattern

The Jacobian of the right hand side of Eq. (5) is

$$\mathbf{J} = \mathbf{A} + \mathbf{B} + \frac{\partial \mathbf{S}}{\partial \mathbf{C}}, \tag{50}$$

which is often needed to improve numerical performance, and knowing its sparsity pattern (i.e., which elements are non-zero) can accelerate numerical computation considerably (Rackauckas and Nie, 2017). The sparsity pattern of $\mathbf{A} + \mathbf{B}$ is set by the discretization scheme, and in SedTrace it is a Julia `Tridiagonal` matrix. The sparsity pattern of $\frac{\partial \mathbf{S}}{\partial \mathbf{C}}$ is model-specific, and

SedTrace detects it during code generation. Without pH and speciation modeling, $\frac{\partial \mathbf{S}}{\partial \mathbf{C}}$ can be treated as a Julia `BandedMatrix`, the upper and lower bandwidths of which are equal to the number of model substance. However, pH and speciation modeling introduces additional coupling to the Jacobian, and increases the bandwidths of $\frac{\partial \mathbf{S}}{\partial \mathbf{C}}$. Thus, it may be better to treat $\frac{\partial \mathbf{S}}{\partial \mathbf{C}}$ as a Julia `SparseMatrixCSC` especially when the size of the Jacobian is large, given that most of the elements on the diagonals of $\frac{\partial \mathbf{S}}{\partial \mathbf{C}}$ are zero.





Non-zero elements can be introduced by direct coupling through kinetic reactions. Such coupling happens within the same grid cell. When parsing the chemical equations in Sect. 4.1.1 SedTrace knows which reactions affect a model substance. And by further parsing the rate expressions of the reactions, SedTrace knows which model substances the reaction rates of these reactions depend on. SedTrace then establishes a dependency relationship for all model substances, and sets the corresponding elements of the Jacobian to non-zero. For example, in `SimpleFe` the summed reaction rate of TFe (`S_TFe`,

see code in Sect. 5.3) includes the reaction `RFeOOHPOC`, the kinetic rate of which depends on FeOOH, O2, and POC. Thus, the $[i_{TFe}+(j-1)M, i_{FeOOH}+(j-1)M]$, $[i_{TFe}+(j-1)M, i_{O2}+(j-1)M]$, and $[i_{TFe}+(j-1)M, i_{POC}+(j-1)M]$ elements of the Jacobian are non-zero. The dependency relationship in the `SimpleFe` model is:

```
Row │ substance   dependence
    │ String      String
─────────────────────────────────────────────
  1 │ H           FeOOH,POC,SO4,TH2S,TCO2,H,TFe
  2 │ POC         FeOOH,POC,SO4
  3 │ FeOOH       FeOOH,POC,TH2S
  4 │ TCO2        FeOOH,POC,SO4
5 │ TFe        FeOOH,POC,TH2S,TCO2,H,TFe,SO4
  6 │ SO4         SO4,FeOOH,POC
  7 │ TH2S        SO4,FeOOH,POC,TH2S,TCO2,H,TFe
  8 │ FeS         TCO2,H,TH2S,TFe,POC,SO4
```

Speciation further introduces transport coupling to the Jacobian that does not happen inside the same grid cell. In the

`SimpleFe` model, the adsorbed Fe concentration depends on the surface POC. Thus transport causes TFe to depend not only on POC inside the same grid cell, but also the two neighboring cells. Therefore the $[i_{TFe}+(j-1)M, i_{POC}+(j-1)M]$, $[i_{TFe}+(j-1)M, i_{POC}+(j-2)M]$, and $[i_{TFe}+(j-1)M, i_{POC}+(j)M]$ elements of the Jacobian are non-zero. SedTrace keeps a record of such relationship during code generation.

        Similar coupling outside the same grid cell is introduced by pH modeling. The proton concentration depends on the

EIs not only inside the same grid cell, but also the two neighboring cells because of coupled transport. Therefore in `SimpleFe` the $[i_{H}+(j-1)M, i_{TCO2}+(j-1)M]$, $[i_{H}+(j-1)M, i_{TCO2}+(j-2)M]$ and $[i_{H}+(j-1)M, i_{TCO2}+(j)M]$ elements of the Jacobian are non-zero. SedTrace knows such dependency relationship internally when generating the pH code.

        SedTrace assembles the non-zero elements and produces the Jacobian sparsity pattern which is saved in `jactype.jl`.

## 6 Numerical solver


        Equation 5 is a system of ODEs that are coupled, nonlinear (in $S$), and stiff. Since TEIs are sensitivity to many biogeochemical processes, a model for TEI diagenesis likely needs to include a large reaction network, together with pH and speciation modeling, which introduces additional nonlinear coupling. And to capture the sharp chemical gradient close to the SWI fine grid is often needed. The number of equations can thus reach greater than $10^4$ as in the case studies presented below.

Our experience shows that the Backward Differential Formula (BDF), a family of implicit linear multistep method of time



stepping, is among the most efficient for solving large systems of stiff diagenetic equations. SedTrace uses the BDF method from the CVODE solver in the SUite of Nonlinear and DIfferential/ALgebraic equation Solvers (SUNDIALS) package (Hindmarsh et al., 2005; Gardner et al., 2022), which is written in C but made accessible to Julia by the Sundials.jl package as part of DifferentialEquations.jl (Rackauckas and Nie, 2017).

At the $n$th time step of integration a system of nonlinear equation resulting from time discretization of Eq. (5) needs to be solved by CVODE (Hindmarsh et al., 2005):

$$\mathbf{F}(\mathbf{C}^n) = \mathbf{C}^n - \gamma_n \mathbf{f}(\mathbf{C}^n) - a_n = 0, \tag{51}$$

where $\mathbf{f}$ is the right hand side of Eq. (5), $\gamma_n$ and $a_n$ are the coefficients set by CVODE. SedTrace solves Eq. (51) using the Newton-Krylov method which is efficient for large sparse systems (Knoll and Keyes, 2004).

The $m$th Newton iteration step to update $\mathbf{C}^n$ involves solving a system of linear equations:

$$(\mathbf{I} - \gamma_n \mathbf{J})\Delta \mathbf{C}^n = -\mathbf{F}(\mathbf{C}^{n,m}), \tag{52}$$

where $\Delta \mathbf{C}^n = \mathbf{C}^{n,m+1} - \mathbf{C}^{n,m}$ is the increment, $\mathbf{I}$ is the identity matrix, $\mathbf{J} = \frac{\partial \mathbf{f}}{\partial \mathbf{C}}$ is the Jacobian of the right hand side of Eq. (5).

The Krylov space iterative method of solving Eq. (52) requires proper preconditioning to be numerically fast (Knoll
and Keyes, 2004). SedTrace applies a right preconditioner by default:

$$((\mathbf{I} - \gamma_n \mathbf{J})\mathbf{P^{-1}})(\mathbf{P}\Delta \mathbf{C}^n) = -\mathbf{F}(\mathbf{C}^{n,m}), \tag{53}$$

SedTrace uses the incomplete LU factorization (ILU) of $\mathbf{I} - \gamma_n \mathbf{J}$ as the preconditioner $\mathbf{P}$. Currently two options are available: the ILU with zero-level fill (ILU0) from the ILUZero.jl (Covalt, 2022) package and the Crout version of ILU with drop tolerance from the IncompleteLU.jl package (Stoppels, 2022). The advantage of ILU0 is memory efficiency. Since the
sparsity pattern of $\mathbf{J}$ does not change during iteration, the sparsity pattern of $\mathbf{P}$ is also fixed. SedTrace can reuse the pre-allocated memory when updating $\mathbf{P}$. In comparison, the ILU with drop tolerance uses more memory because during each iteration the sparsity pattern of $\mathbf{P}$ may change and new memory needs to be allocated, but it has the advantage that the resulting factorization is a better approximation of $\mathbf{I} - \gamma_n \mathbf{J}$ than ILU0.

Creating the preconditioner requires computing $\mathbf{J}$. SedTrace computes $\mathbf{J}$ using the forward mode automatic
differentiation tools from ForwardDiff.jl (Revels et al., 2016). This computation is accelerated using matrix coloring algorithm (Gebremedhin et al., 2005) from SparseDiffTools.jl (Gowda et al., 2022) taking advantage of the knowledge of the sparsity pattern detected by SedTrace in Sect. 5.4. The preconditioned system Eq. (53) is then solved using an iterative method, for example the generalized minimal residual method (GMRES).

## 7 Model simulation and output

The user generates model code and performs simulation in the `main.jl` file. The example of `SimpleFe` is shown here:





```
      using SedTrace

      # model configuration
modeldirectory = (@__DIR__) * "\\"
      modelfile = "model_config.SimpleFe.xlsx"
      modelname = "SimpleFe"
      modelconfig = ModelConfig(modeldirectory, modelfile, modelname)

# generate a parameter sheet template
      @time generate_parameter_template(modelconfig)
      # generate model code
      @time generate_code(modelconfig)
      # load model code files
IncludeFiles(modelconfig)

      # initial values
      C0 = Param.C0;
      # initialize parameters
parm = Param.ParamStruct();
      # initialize cache and ODE function
      OdeFun = Cache.init(C0, parm.Ngrid);
      # initialize Jacobian
      JacPrototype = JacType(Param.IDdict);

      # test if the Jacobian is correct
      TestJacobian(JacPrototype, OdeFun, parm)
      # benchmark the ODE function performance
      BenchmarkReactran(OdeFun, C0, parm)
# benchmark the Jacobian performance
      BenchmarkJacobian(JacPrototype, OdeFun, parm)
      # benchmark the preconditioner performance
      BenchmarkPreconditioner(JacPrototype, OdeFun, parm,:ILU0)

# configure the solver
      solverconfig = SolverConfig(:GMRES, :ILU0, 2)

      # configure the solution
      solutionconfig = SolutionConfig(
C0, # inital values
         (0.0, 3000.0), # time span
         reltol = 1e-6, # relative tolerance
         abstol = 1e-18, # absolute tolerance
         saveat = 100.0, # save time steps
callback = TerminateSteadyState(1e-16, 1e-6),
         # terminate when steady state is reached
      );

      # run the model
solution = modelrun(OdeFun, parm, JacPrototype, solverconfig, solutionconfig);

      # generate output and plot
      generate_output(modelconfig, solution, showplt = true,saveplt=true)
```

The workflow (Fig. 1) begins with configuring the model using `modelconfig`, supplying information of the model

directory, the excel file and model name. If needed the user can call

`generate_parameter_template(modelconfig)`, which parses the `substances`, `reactions`, `speciation`





and `adsorption` sheets to identify which parameters are needed by the model and output a template `model_parameter_template.xlsx` to assist the creation of the `parameters` sheet. Once the Excel model configuration file is created, code generation is done by calling `generate_code(modelconfig)`, creating the 5 Julia files discussed in Sect. 5. The Julia files needs to be loaded into the workspace `module Main` by calling `IncludeFiles(modelconfig)`. The parameters are loaded into the `module Param`, and the cache and ODE function are loaded into the `module Cache`. During code generation, SedTrace collects the results of parsing the Excel sheets and creates a file `model_parsing_diagnostics.xlsx`, which can help the user to diagnose potential issues of code generation.

In the next step the user initializes the initial values, parameters, ODE function and Jacobian pattern. Internally SedTrace generates a set of initial values `C0` that are constant with respect to depth based on user supplied boundary conditions (Meysman et al., 2003). The user can also supply their own initial values, for example using the output from previous model runs. The parameters are initialized by `parm = Param.ParamStruct()`. The ODE function is initialized using `OdeFun = Cache.init(C0, Ngrid)`. The Jacobian sparsity pattern is initialized by `JacPrototype = JacType(Param.IDdict)`, where `IDdict` is a Julia `Dictionary` that stores the indices of the model substances.

SedTrace also provides functions for code testing. Function `TestJacobian()` computes the Jacobian assuming it is dense, which is time consuming but accurate. The result is then compared with the Jacobian computed using `JacPrototype`. This serves as a check on code generation. `BenchmarkJacobian()`, `BenchmarkReactran()`, `BenchmarkPreconditioner()` are used to benchmark the performance and memory allocations of the Jacobian, ODE and preconditioner functions respectively.

The user configures the numerical solver using `solverconfig = SolverConfig(:method, :preconditioner, prec_side)`, where `:method` is the numerical method such as `:GMRES`, `:preconditioner` is the name of the preconditioner, by default `:ILU0`, and `prec_side` controls whether it is left (1) or right (2) preconditioning in Eq. (53). The numerical solution is configured using `solutionconfig = SolutionConfig(C0, tspan, reltol, abstol; callback)` to set the initial values (`C0`), the time span (`tspan`), and the relative (`reltol`) and absolute (`abstol`) tolerances for numerical convergence. Any `callback` function compatible with DifferentialEquations.jl can be supplied too. For example, the user can use `TerminateSteadyState(rtol, atol)` from the DiffEqCallbacks.jl (Rackauckas and Nie, 2017) library to terminate the simulation once steady-state is reached given the relative and absolute tolerances of $\frac{d}{dt}\mathbf{C}$ (`rtol` and `atol`).

Model simulation is performed by calling `solution = modelrun(OdeFun, parm, JacPrototype, solverconfig ,solutionconfig)`. Internally SedTrace creates the Jacobian, solver and preconditioner functions and format the ODE function to be compatible with DifferentialEquations.jl, which carries out the numerical solution.





Finally, outputs are created by calling `generate_output(modelconfig, solution; site = nothing, showplt = true, saveplt = false)`. SedTrace will compute the output variables listed in the `output` sheet (Table 7). New variables can be computed by supplying their mathematical expressions as functions of the model substances in the `expression` column. For example, if the user wants to output pH on the free proton scale, then an expression `-log10(H)` should be supplied. SedTrace also converts the default units to units specified by the user in the `unit_profile` column, by multiplying the conversion factors in the `conversion_profile` column. SedTrace can also computes the benthic fluxes of the output variables at the SWI. This is enables by set the `flux_top` column to 1. Similar to the model profiles, unit conversion for the flux is done using the `conversion_flux` and `unit_flux` columns.

SedTrace then plots the profiles and the fluxes of the output variables. The user can supply the measured profiles of these variables in the `data` sheet and the measured fluxes in the `flux_top_measured` column in the `output` sheet. SedTrace will plot the measurements with the model output. To do so the user needs to specify the site name in the `site` column and supply this name to `generate_output`. The name of the `substance` and the `unit` in the `data` sheet must match exactly those in the `output` sheet for SedTrace to match the model results with measurements. Measured data are supplied to the `depth` and `value` columns, with optional `error` values that will be used to create error bars on the plots. SedTrace will save the output in `model_output.xlsx`, containing internal output of modeled substances (sheet `Substances`), reaction rates (`ReacRates`), saturation state (`Omega`), pH related species (`pHspecies`), speciation results (`Speciation`) and intermediate variables (`IntermediateVar`) in the default SedTrace units. The user specified output is reported in the `OutputProfile` and `OutputFlux` sheets with the custom units. If `saveplt = true` SedTrace will save the plots in a `plots` directory inside `modeldirectory`.

## 8 Case studies

In the `/examples` folder we provide case studies of the generation and running of diagenetic models at different levels of complexity. These examples include models with analytical solutions that are used to validate SedTrace's code generation and numerical methods. We also present advanced cases studies to demonstrate SedTrace's capacity for modeling pH, speciation, radiogenic and stable isotopes.

### 8.1 Models with analytical solutions

A few simple models of the diagenesis of carbon and nutrients that have analytical solutions are presented in `examples/analytical`. These include the `POC1G` model for POC remineralization using the 1-G kinetics, the `Ammonia` model for organic N remineralization and $NH_4^+$ adsorption, the `SulfateRedcution` model for oxidation of POC by sulfate, and the `Phosphate` model for organic P remineralization and authigenic phosphate precipitation. These examples come from Berner's classic textbook (Berner, 1980).



Here we discuss in more detail the model `pHBB1991`. Boudreau (1991) created a diagenetic model with an analytical

solution to explain the pH change across the mat of sulfide oxidizing bacteria *Beggiatoa* in sediments from the Danish lagoons

(Jørgensen and Revsbech, 1983). This model is now generated here using SedTrace. It includes one kinetic reaction, the

oxidation of HS⁻ by $O_2$. The kinetic rate is $k_{OS}e^{-a(x-x_0)^2}$, where $k_{OS}$ is the rate constant, $x$ is depth. The reaction is assumed

to happen close to the mat at $x_0 = 0.005$ cm where dissolved $O_2$ disappears and $H_2S$ starts to increase, and $a$ controls the

sharpness of this interface. The model substances are dissolved $O_2$, $H^+$ and the EIs $TCO_2$, $TH_2S$ and $TH_3BO_3$. Their Dirichlet

boundary conditions are specified at the top (-0.05 cm) and bottom (0.15 cm) of the model domain. Porosity is assumed to be

constant and equal to 1, and thus no distinction is made between seawater above the SWI and the pore water below. Molecular

diffusion is the only transport mechanism.

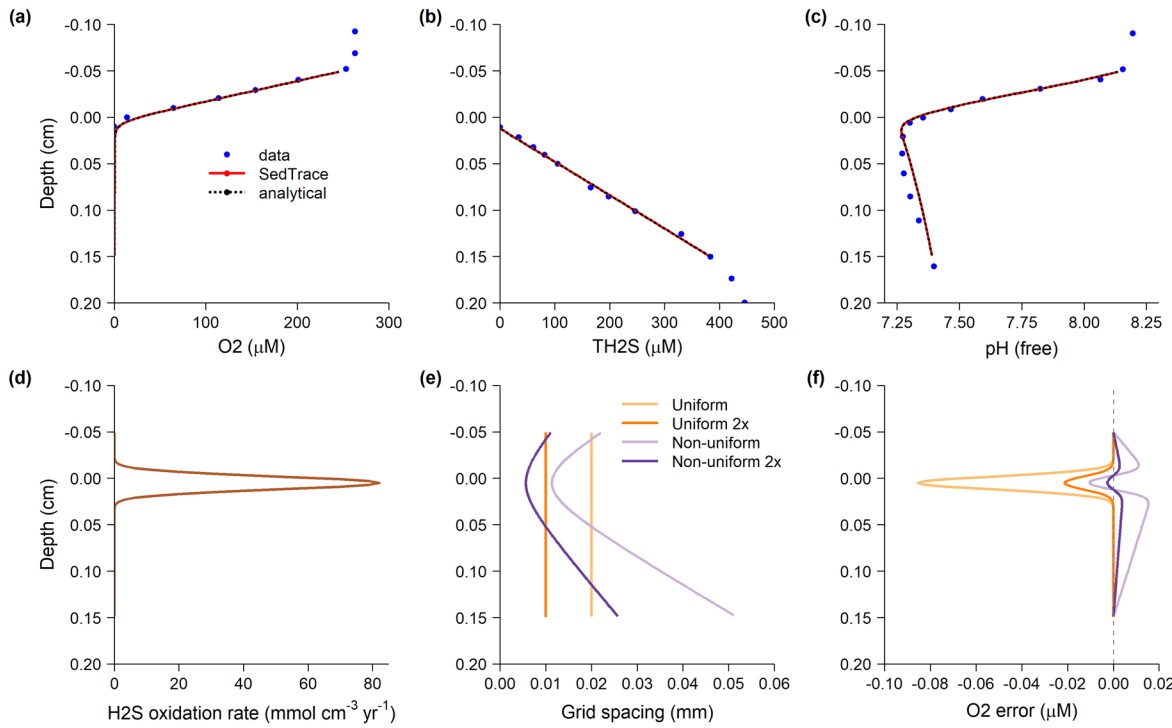

**Figure 2. Results of the `pHBB1991` model compared with analytical solution (Boudreau, 1991) and observations (Jørgensen and Revsbech, 1983).** (a) Dissolved $O_2$. (b) Dissolved total $H_2S$. (c) pH on the free proton scale. (d) Rate of $H_2S$ oxidation by $O_2$. In (a) to (d)
the model solutions are computed using the uniform grid with `Ngrid = 100`. (e) Model grid cell size. (f) Errors of modeled $O_2$ concentration, defined as the difference between the model solution and the analytical solution. In (e) and (f) we show the model solutions on four grids: the uniform grid with `Ngrid = 100` and `Ngrid = 100` (2x), the non-uniform grid with `Ngrid = 100` and `Ngrid = 200` (2x).

We tested the sensitivity of the model error (with respect to the analytical solution) to the model grid. We created the

1000 model with either a uniform or a non-uniform grid. The `gridtran` for the uniform grid is simply `x->x-0.05`, where the

0.05 cm offset is to take into account the fact that SedTrace's internally uniform grid starts from 0 cm. The `gridtran` for the

non-uniform grid is constructed using hyperbolic functions (Hoffmann and Chiang, 2000):



$$gridtran(x) = (x_0 + 0.05)\left(1 + \frac{\sinh\left(b(\frac{x}{L}-A)\right)}{\sinh(bA)}\right) - 0.05, A = \frac{1}{2b}\ln\frac{1+(e^b-1)(x_0+0.05)/L}{1+(e^{-b}-1)(x_0+0.05)/L}, \tag{54}$$

where $L = 0.2$ cm is the length of the model domain and 0.05 cm is the offset. The resulting grid points are concentrated near $x_0 = 0.005$ cm, the degree of which is controlled by $b$. We tested both types of grid with `Ngrid` of 100 and 200.

The model results, the analytical solution and the observations made using profiling micro-electrodes, are shown in Fig. 2. The non-uniform grid captures the sharp biogeochemical gradient near $x_0 = 0.005$ cm better than the uniform grid when `Ngrid` is the same. The $L^\infty$ norm of the model errors of $O_2$ are 0.086, 0.021, 0.015 and 0.0039 µM for the uniform grid (`Ngrid=100`), uniform grid 2X (`Ngrid=200`), non-uniform grid (`Ngrid=100`), and uniform grid 2X (`Ngrid=200`) respectively. Thus using a suitable grid can considerably improve the model accuracy.

**8.2 Oregon margin diagenetic Nd cycle**

Neodymium is one of the Rare Earth Elements (REE), which are important tracers in chemical oceanography (Elderfield and Greaves, 1982). Its radiogenic isotope composition, expressed as $\varepsilon_{Nd} = \left(\frac{^{143}Nd/^{144}Nd_{Sample}}{^{143}Nd/^{144}Nd_{CHUR}} - 1\right) \times 10^4$ where CHUR is the chondritic uniform reservoir, has been used to study modern and past ocean circulation, marine and continental weathering (Goldstein and Hemming, 2003; Haley et al., 2017; Lacan and Jeandel, 2005; Frank, 2002). It is designated as a "key parameter" by the GEOTRACES program, so that it needs to be measured by all affiliated cruises. One of the greatest challenge facing the study of the modern ocean Nd cycle is that its sedimentary cycle is poorly constrained. Recently studies suggest that a benthic flux from marine sediments, particularly in the deep sea, is likely the dominant source of seawater Nd, far exceeding that of riverine and dust inputs at the ocean surface (Abbott et al., 2015b; Du et al., 2018, 2020; Haley et al., 2017). Consequently, pore water $\varepsilon_{Nd}$, subject to diagenetic processes such as marine silicate weathering, can affect the water column $\varepsilon_{Nd}$ (Abbott et al., 2015a, 2016; Du et al., 2016). Because of low Nd concentration, ~1 L of pore water is typically required to make an isotope measurement. Thus pore water $\varepsilon_{Nd}$ analysis to date has only been done at three sites on the Oregon margin in the Northeast Pacific (Abbott et al., 2016, 2015a). Modeling is needed to make efficient use of the data and extrapolate to other regions once the diagenetic processes are well understood.

We recently published a reactive-transport model for the early diagenesis of Nd at the deep sea site (3000 m water depth) on the Oregon margin (Du et al., 2022), which is regenerated here using SedTrace. This model has 41 kinetic reactions, including the organic matter remineralization sequence, secondary redox reactions, and the diagenesis of carbonate, sulfide and opal. Pore water pH is modeled by including the EIs TCO2, TH2S and TH3BO3. Aqueous speciation of $Fe^{2+}$ and $Al^{3+}$ are included to model mineral dissolution/precipitation. Adsorption of $Fe^{2+}$ and $Mn^{2+}$ onto Fe/Mn oxides are also included. $^{144}Nd$ (`Ndnr`) and the radiogenic $^{143}Nd$ (`Ndr`) are modeled as two tracers. The following aqueous and solid phase speciation are included for both isotopes: complexation with $CO_3^{2-}$, $HCO_3^-$, $OH^-$, $Cl^-$, $SO_4^{2-}$ and $H_3SiO_4^-$, adsorption onto Fe and Mn oxides, incorporation into Fe/Mn oxides by co-precipitation, formation of the authigenic phosphate mineral rhabdophane, released as trace constituents from primary silicates. By including co-cycling with Fe, Mn and phosphate, and release by marine silicate

weathering coupled to reverse weathering, the model successfully simulated the pore water Nd concentration and $\varepsilon_{Nd}$ data (Fig.

3). This model is relatively large, containing 10,200 equations of 34 model substances on 300 grids between 0 cm and 50 cm. Interested readers should consult Du et al., (2022) for details of model description. This model can be found in `/examples/OregonNd`.

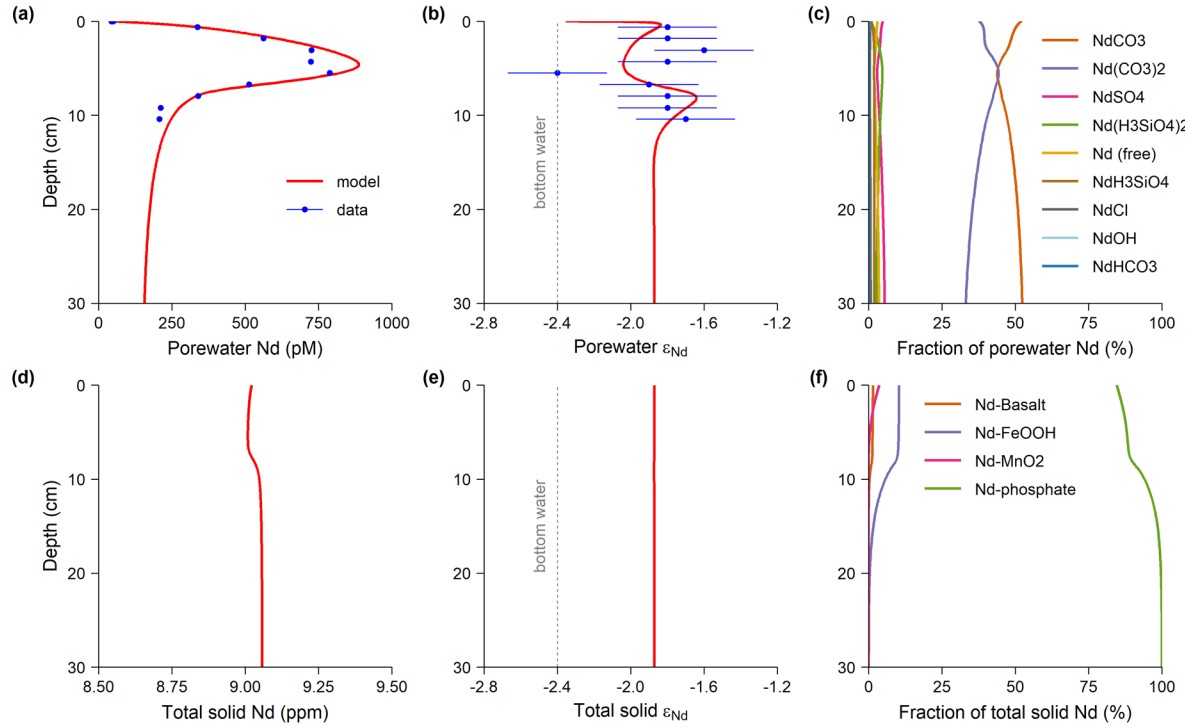

**Figure 3. Results of the `OregonNd` model compared with measurements (Du et al., 2022; Abbott et al., 2015a). (a)** Pore water Nd
concentration. **(b)** Pore water $\varepsilon_{Nd}$. Error bars of the data are 2 standard deviations. **(c)** Pore water Nd speciation. **(d)** Total solid Nd, including authigenic Nd associated with Fe/Mn oxide and phosphate, and lithogenic Nd associated with basalt. This does not include other lithogenic Nd not in the model. **(e)** Total solid $\varepsilon_{Nd}$. **(f)** Solid Nd speciation.

### 8.3 Santa Barbara Basin sediment biogeochemistry, pH and Mo

   Santa Barbara Basin (SBB) is one of the California borderland basins. Its seasonally anoxic condition leads to organic
rich and laminated sediments (Reimers et al., 1996). SBB is among the most studied location for sediment diagenesis, and has perhaps one of the most complete pore water dataset to offer in the literature (Reimers et al., 1996). High quality pH measurements by *in situ* profiling micro-electrode, and the availability of various TEI data make it ideal for benchmarking diagenetic models (Meysman et al., 2003). Using SedTrace, we generated a diagenetic model for SBB that includes sediment biogeochemistry, pH and Mo cycling. This example is included in the `/examples/SBB`.

The biogeochemical reaction network includes the classic redox sequence of aerobic respiration, denitrification, Mn and Fe reduction, sulfate reduction and methanogenesis. The model also includes secondary redox reactions, and the

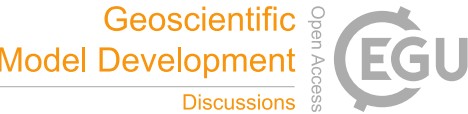

authigenesis of carbonates, sulfide, opal and carbonate fluorapatite (CFA). The adsorption of $NH_4^+$, $Fe^{2+}$ and $Mn^{2+}$ are treated using the linear isothermal. To model pH, we include the following equilibrium invariants: $TCO_2$, $TH_2S$, $THSO_4$, $TH_3BO_3$, $TH_3PO_4$ and THF. The biogeochemical model includes 26 substances and 36 kinetic reactions. We use a 500-point non-uniform grid with finer spacing close to the SWI. The total number of equations in the biogeochemical model is thus 13,000. Figure 4 shows the modeled sediment biogeochemistry.



**Figure 4: Santa Barbara Basin sediment biogeochemistry.** Concentrations of solid sediment substances: (**a**) Particulate organic carbon, (**b**) Mn oxide, (**c**) Fe oxide, (**d**) Particulate inorganic carbon, (**e**) FeS, (**f**) pyrite. Concentrations of dissolved substances: (**g**) $O_2$, (**h**) $NO_3$, (**i**) $NO_2$, (**j**) Mn, (**k**) Fe, (**l**) Ca, (**m**) $NH_4$, (**n**) total phosphate, (**o**) total dissolved inorganic carbon, (**p**) total alkalinity, (**q**) total sulfate, (**r**) total sulfide, (**s**) total fluoride, (**t**) total boron. Names that with prefix "T" (e.g., $TCO_2$) indicate equilibrium invariants. Data are from Reimers et al., (1996).

The top 50 cm of the sediment represents ~150 year of sedimentation. The model successfully reproduced the measurements of pore water constituents, which respond much faster to perturbations of the seasonal cycle and other variabilities, while the solid sediment components show non-steady state behaviors (Fig. 4). During the sampling time bottom



water was low (~9 μM) but not anoxic, and the oxygen penetration depth was ~1 cm, below which $H_2S$ is immediately detectable. There was active formation of authigenic minerals (Reimers et al., 1996). Intense Fe cycling leads to high concentrations of FeS and $FeS_2$. Decreasing pore water Ca concentration is evidence of authigenic carbonate precipitation, and the model shows that Ca-, Fe- and Mn- carbonates are likely formed. Decreasing pore water F concentration, and relatively

low $PO_4$ concentration, are explained by CFA precipitation in the model (Jahnke, 1984; Reimers et al., 1996).

The model also captures the measured pH profile (Fig. 5). The pH measurements were reported on the seawater scale, and the model computes pH on this scale by summing the concentrations of free proton, $HSO_4^-$ and HF. Using DSA (Hofmann et al., 2008), the model can easily partition the changes of pH to relevant transport and reaction terms (Fig. 5b–e). The overall increase in pH is driven by Fe reduction in the model (Fig. 5d), as suggested by previous studies (Reimers et al., 1996;

Boudreau and Canfield, 1988). This increase in pH is responsible for the saturation and precipitation of authigenic carbonate and CFA. The slight decrease of pH at ~3.5 cm (Fig. 5a) is modeled by FeS precipitation which releases protons to pore water (Fig. 5d). The reaction rates of proton is balanced by the transport rate largely attributed to by total alkalinity, $TCO_2$ and $TH_2S$ (Fig. 5b).

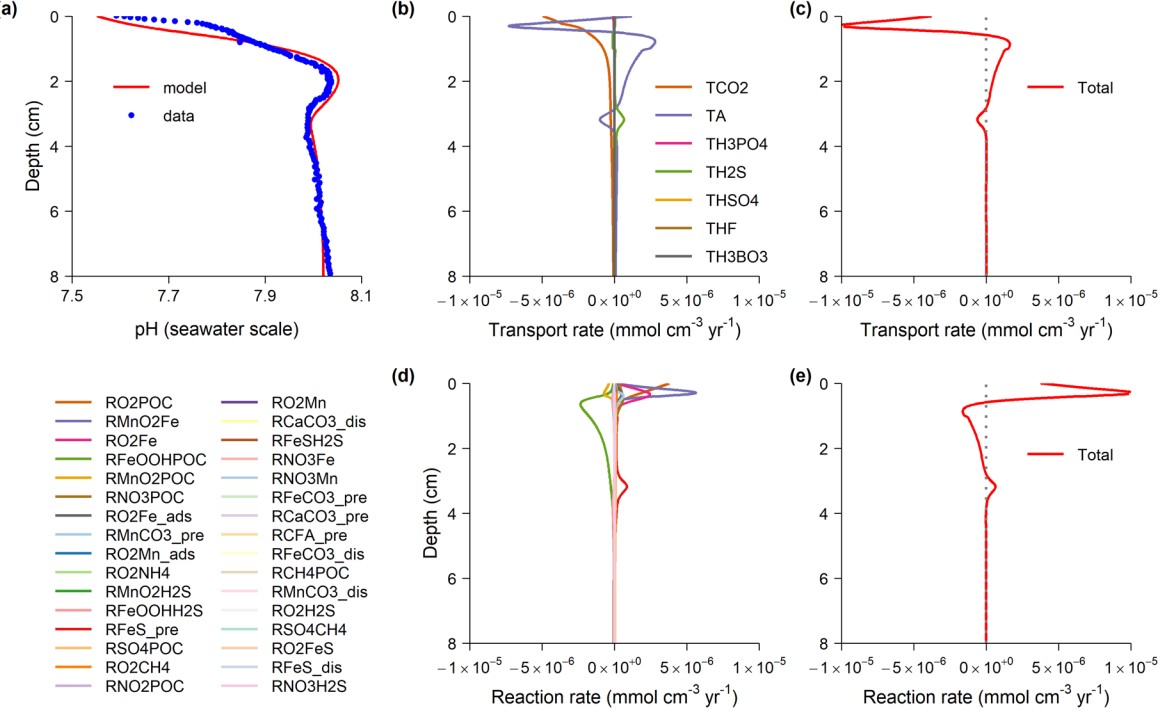

**Figure 5: Santa Barbara Basin pore water pH. (a)** Modeled pH compared with measured values using *in situ* profiling micro-electrode probe (Reimers et al., 1996). **(b)** Rate of $[H^+]$ change due to the transport of the equilibrium invariants and total alkalinity. **(c)** The summed transport rate of $[H^+]$. **(d)** Rate of $[H^+]$ change due to the biogeochemical reactions. Legends is plotted on the left side. Reactions names indicate either redox reactions, such as "RO2POC" in the format of oxidant followed reductant, or mineral dissolution/precipitation reactions with suffix "dis/pre". **(e)** The summed reaction rate of $[H^+]$.





Molybdenum is sensitive to sedimentary redox condition, and its stable isotope composition, expressed as

$\delta^{98}Mo = (\frac{^{98}Mo/^{95}Mo_{sample}}{^{98}Mo/^{95}Mo_{standard}} - 1) \times 10^3 + 0.25$, where NIST SRM-3134 is the commonly used standard and its $\delta^{98}$Mo is

0.25 ‰ by convention, is an important proxy to study past ocean deoxygenation (Kendall et al., 2017). SBB provides a useful

analogy for an anoxic ocean, and modeling the sedimentary Mo cycle here may help understand how the $\delta^{98}$Mo works as a

redox proxy. Here we present a *test* model for Mo diagenesis in SBB, to demonstrate the capability of SedTrace for modeling

stable isotope fractionation, complementing the radiogenic isotope example above.

In this model, we consider 5 dissolved Mo species, $MoO_4^{2-}$ and 4 thiomolybdate species (Erickson and Helz, 2000):

$$MoO_4^{2-} + iH_2S = MoO_{4-i}S_i^{2-} + iH_2O \ (i = 1 \ to \ 4), \ K_i = \frac{[MoO_{4-i}S_i^{2-}]}{[MoO_4^{2-}][H_2S]^i}, \tag{55}$$

$K_i$ are the apparent equilibrium constants.

We include $^{98}$Mo (Moh) and $^{95}$Mo (Mol) as two tracers. We assume that equilibrium isotope fractionation is induced

during thiolation:

$$\alpha_i^{98/95} = \frac{^{98}MoO_4^{2-}}{^{95}MoO_4^{2-}} / \frac{^{98}MoO_{4-i}S_i^{2-}}{^{95}MoO_{4-i}S_i^{2-}} = K_i^{98}/K_i^{95}, \tag{56}$$

$\alpha_i^{98/95}$ are the fractionation factor, which are 1.0014, 1.0028, 1.00455 and 1.0063 for $i = 1$ to 4 respectively estimated

by *ab initio* calculation (Tossell, 2005) and recalculated by Kendall et al., (2017). In SedTrace, we add Eq. (55) to the

speciation sheet, choosing $MoO_4^{2-}$ as the base species. Presently SedTrace does not provide special treatment of isotope

fractionation, so the user needs to incorporate the fractionation factor in the parameters (e.g., $K_i$) before supplying them to

SedTrace. We use the constants from Erickson and Helz (2000) as $K_i^{95}$ and then multiply them by $\alpha_i^{98/95}$ to get $K_i^{98}$.

We test the model sensitivity to the Mo removal mechanism. In Case 1, we assume all thiomolybdate species can be

removed by scavenging:

$$RMo_{rm1} = k_{rm1} \sum_i [MoO_{4-i}S_i^{2-}]. \tag{57}$$

In Case 2 we assume that only tetrathiomolybdate can be removed by scavenging:

$$RMo_{rm2} = k_{rm2}[MoS_4^{2-}]. \tag{58}$$

In this test we do not consider kinetic isotope fractionation during removal and diffusion, and reactions Eq. (55) are

assumed to be fast enough to reach local equilibrium. In Eq. (57) we assume all species are removed at the same rate constant.

We assume the bottom water $\delta^{98}$Mo to be the same as the seawater, which is globally uniform at 2.34 ‰ (Kendall et al., 2017).

The only source of authigenic Mo accumulation is pore water Mo removal supported by diffusion of seawater into sediment.

In the model we also supply a lithogenic Mo flux to account for the reported 2 ppm lithogenic Mo in sediments (Zheng et al.,

2000). The lithogenic $\delta^{98}$Mo is assumed to be the same as the Upper Continental Crust (UCC) ~0.3 ‰ (Kendall et al., 2017).

In SedTrace we further add 6 Mo tracers to the biogeochemical model described above, including the two Mo isotopes

in pore water, lithogenic and authigenic fractions, increasing the total number of model equations to 16,000.



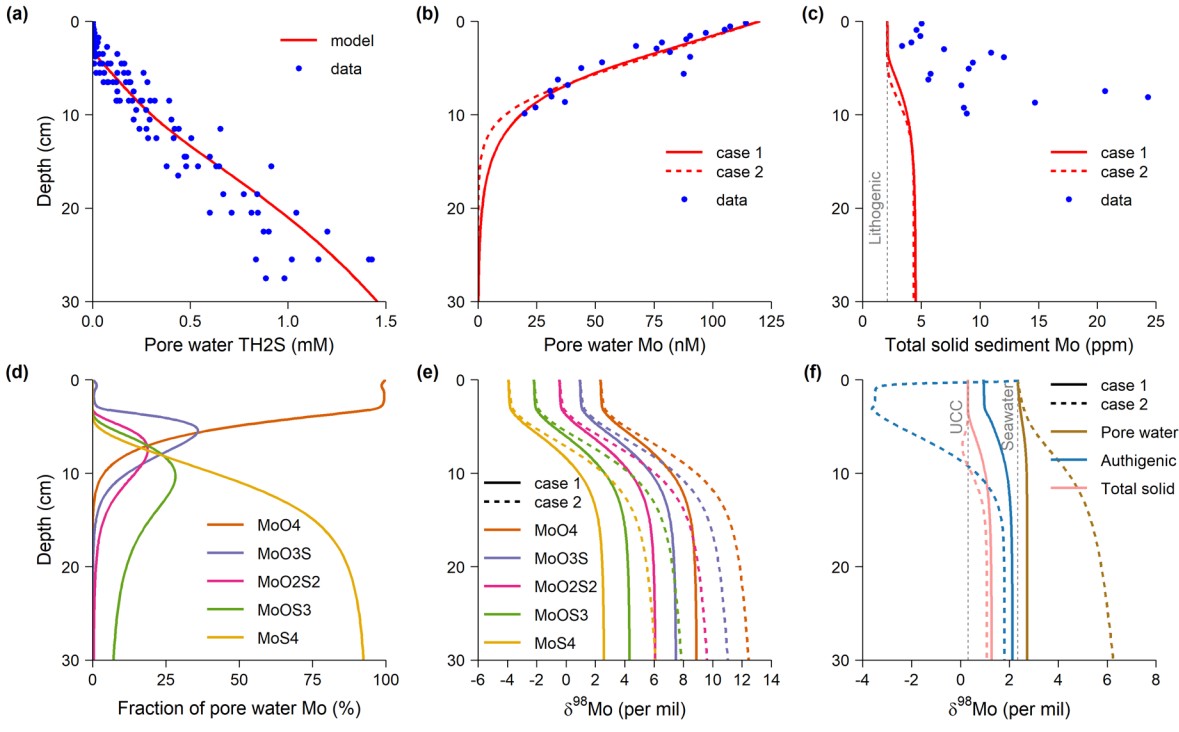


**Figure 6: Santa Barbara Basin sediment Mo cycle.** (**a**) Modeled and measured total sulfide (Reimers et al., 1996). (**b**) Modeled and measured pore water Mo (Zheng et al., 2000). (**c**) Modeled and measured total solid sediment Mo (Zheng et al., 2000). (**d**) Modeled pore water Mo speciation. (**e**) Modeled $\delta^{98}$Mo of the dissolved species. (**f**) Modeled $\delta^{98}$Mo of pore water, authigenic and total solid sediment.

In the two model cases we vary $k_{rm1}$ and $k_{rm2}$ to fit the measured pore water Mo concentration profile (Fig. 6b)

(Zheng et al., 2000). The model results of the two cases are roughly the same. As H$_2$S concentration increases, pore water Mo speciation is dominated by the thiomolybdate species (Fig. 6d). The aqueous speciation in the two cases is identical since it only depends on the concentration of H$_2$S. The modeled total solid (sum of authigenic and lithogenic) sediment Mo concentrations in the two cases are also similar, but both are much lower than measured (Fig. 6c) (Zheng et al., 2000). Modeled authigenic Mo enrichments, relative to the 2 ppm lithogenic background, do not start until below ~5 cm. The model results

suggest that the Mo diffusion and removal rate at the time of pore water sampling cannot explain the history of sediment Mo enrichment in the top 10 cm. Non-steady state accumulation, and other Mo sources, such as Mo carried into sediments by settling particles, are needed to close the sedimentary budget (Zheng et al., 2000).

Modeled thiomolybdate species all have lighter $\delta^{98}$Mo than MoO$_4^{2-}$ (Fig. 6e). The removal of thiomolybdate thus makes the residual pore water $\delta^{98}$Mo heavier than seawater (Fig. 6f). In Case 1 where all thiomolybdate species are removed,

the *apparent* fractionation between pore water and authigenic sediment (<1.4 ‰) is much smaller than in Case 2 where only tetrathiomolybdate is removed (as much as 6.5 ‰). In contrast, the $\delta^{98}$Mo of total sediment has little sensitivity to the removal mechanism (Fig. 6f). This illustrates the under-appreciated challenge of applying detrital corrections to TEI data: the most





useful data to differentiate the removal mechanism in this test is the authigenic $\delta^{98}$Mo at near zero authigenic enrichment, which unfortunately will have the greatest uncertainty when applying detrital correction in reality (Ciscato et al., 2018).

As pore water Mo removal becomes quantitative below 25 cm, authigenic $\delta^{98}$Mo approaches the seawater $\delta^{98}$Mo. However, unlike in a closed system, modeled authigenic $\delta^{98}$Mo never reaches the seawater value even at depths of quantitative removal. This is because in a reactive-transport system, the memory of the authigenic enrichment of light Mo isotope in the zone of partial removal will persist.

## 9 Future developments

Future developments of SedTrace will improve the user interface, the speciation modelling capacity and parameter selection. Currently SedTrace has no specialized interface for isotopes, and the user needs to add the isotopes as individual tracers and supply parameters that already incorporate stable isotope fractionation factor and radiogenic isotope source. A more user friendly interface for modeling isotopes is planned. Also, the current implementation of adsorption and aqueous speciation can be further improved to allow more flexible and mechanistic formulations of speciation, for example, including

the speciation of dissolved and surface ligands. At the moment, SedTrace only precomputes selected parameters such as diffusion coefficients and acid dissociation constants while requires the user to supply the rest. Integration with the Miami ion interaction model for seawater (Pierrot and Millero, 2017) and the Pitzer equation based seawater speciation model being developed by the SCOR working group (Humphreys et al., 2022) would allow SedTrace to precompute more parameters and lessen the burden on the user.

Future developments will also aim to add more choices of numerical methods and improve numerical performance. Currently SedTrace only uses CVODE with the iterative method of solving linear systems (Hindmarsh et al., 2005; Gardner et al., 2022). Although we have found it to be more efficient when solving large stiff system of equations, in other cases other ODE solvers or the direct method of solving linear systems may be preferable. Further release will enhance the integration of SedTrace with DifferentialEquations.jl (Rackauckas and Nie, 2017) and provide the user with more numerical choices.

Similarly, more options of preconditioners can also be added. Also, SedTrace currently has no parallel computing capacity (aside from those done internally by Julia), and model simulation cannot take advantage of the multicore architecture of modern computers. Future development will focus on enabling SedTrace for parallel computing, for example, through integration with the PETSc.jl package, the Julia interface for the Portable, Extensible Toolkit for Scientific Computation system that uses the Messaging Passing Interface (MPI) for scalable solution of differential equations (Balay et al., 2022). Such high performance

is needed if SedTrace is to be used to solve global scale problems.

       Currently SedTrace's capacity is limited to the forward modeling of steady-state diagenesis. Special interface can be added to enable the user to add time-dependent forcing in parameters or boundary conditions for fully transient modeling. Moreover, since SedTrace already enables automatic differentiation, capacities of adjoint sensitivity analysis, parameter





estimation, inverse modeling and data assimilation can be added by integration with relevant existing Julia packages
(Rackauckas and Nie, 2017; Dunning et al., 2017).

**Code availability**

Code of SedTrace 1.0, and the excel sheets and Julia scripts used in the case studies, can be found online at the GitHub repository (https://github.com/JianghuiDu/SedTrace.jl) and the Zenodo repository (https://doi.org/10.5281/zenodo.7225861) (Du, 2022).

**Data availability**

No original data is generated by this study. Source data of the pore water and sediment observations used for model evaluation in the case studies can be found in their original references as cited in the main text, and also contained in the `SedTrace/examples/` case study directory (https://doi.org/10.5281/zenodo.7225861) (Du, 2022).

**Author contributions**

JD designed and coded SedTrace. JD performed the model simulations in the case studies and wrote the manuscript.

**Competing interests**

The authors declare that they have no conflict of interest.

**Acknowledgements**

We thank Derek Vance for suggestions that improved the manuscript. We thank Clare Reimers for sharing the data from the
Santa Barbara Basin. SedTrace was inspired by many published diagenetic models, including CANDI (Boudreau, 1996), STEADYSED (Wang and Van Cappellen, 1996), MEDIA (Meysman et al., 2003) and ReacTran (Soetaert and Meysman, 2012).

**Financial support**

This project has received funding from the European Union's Horizon 2020 research and innovation programme under the
Marie Skłodowska-Curie grant agreement 891489. This work was supported by an ETH Zurich Postdoctoral Fellowship 19-2 FEL-32.



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
