# Peer review of "SedTrace 1.0: a Julia-based framework for generating and running reactive-transport models of marine sediment diagenesis specializing in trace elements and isotopes"

_Geoscientific Model Development, 2022_

## Referee Comment (RC2)

It was a pleasure to read through and learn about the new model Sed Trace, in the paper **"SedTrace 1.0: a Julia-based framework for generating and running reactive-transport models of marine sediment diagenesis specializing in trace elements and isotopes"** by Jianghui Du.

The model being described is a 1D (pseudo-)steady-state implementation of a sediment early diagenesis model, that is able to include carbon and nutrients, trace elements, and isotopes. This type of model is notoriously difficult to setup, apply and master as users must have a good command of numerical methods, and many sub-disciplines of (computational) geochemistry, and experience with approaches to validate and assess. Yet, the model scope is broadly relevant to many topical areas of geoscience, and developing new methods and tools to reduce the barrier to entry is of great interest to the community. The paragraph starting line 62 may benefit from the paper by Paraska et al (2014) which similarly highlights the challenge after doing a meta-analysis of models.

As suggested in the title, the software developed in this case is quite unique, as the user can setup a tailored simulation relatively simply in an xlsx spreadsheet, and then this information is parsed to create full-featured simulation in the Julia language. Outputs are similarly written to xlsx for subsequent plotting/analysis. As someone who has spent much time wrestling with fortran versions of similar models (including CANDI), I found the idea quite clever and from a user experience point of view the approach of having a lot of complex configuration information nicely summarised in a xls spreadsheet was a very clean and flexible way to do this.

At the heart of the model is an implementation of a reactive-transport code following the traditional approaches introduced in the foundational models of CANDI, STEADYSED etc, and the numerical methods for discretisation and ODE solution seem closest to CANDI, though updated (e.g. using the robust CVODE). In my view, it is elegantly described and I found it hard to fault, with care and attention to detail in the equations.

The model allows for a grid transformation (e.g. changing resolution over depth). The opening of section 3.1 may benefit from an improved opening/contextual statement setting the scene for this; whilst experienced users will get this concept, it might be confusing new users unclear what they should do or consider when making a grid and options available. Also, our experience with CANDI numerical code was that user's need to be aware of setting the grid parameters, and in some cases they would lose some mass conservation – this was albeit in more unsteady conditions, but this made me wonder if mass balance testing had been done on this code, or could be reported to the user to check for grid issues?

The model includes options for both kinetic and equilibrium chemical reactions, and is quite flexible in allowing users to formulate their reactions and their stoichiometry. The parser is able to allow users to easily tailor their settings and reaction network in Excel rather than having to edit code, and includes options to check stoichiometric balance. The approach to allow for Omega on reactions allows for precipitation/dissolution, and the code adopts a

logistic rather than Heaviside style to smooth the numerical solution when quantities are close to equilibrium.

The pH solution is also well implemented, essentially conserving charge by adopting the concept of the equilibrium invariant concentration (similar to a component definition in PHREEQC). This can be a headache when dealing with boundary conditions, but this is addressed in the model. A comment here how this relates to or is different from traditionally used approaches may be warranted.

Overall, I found the model description excellent, and the software is full-featured and should be published. My main comment is that there is a fair amount of assumed knowledge required to digest these options and methods, and in the spirit of making this type of modelling easier for a wider audience, then the paper may also benefit from some careful additions of some leading or contextualising sentences before diving into some of the detail detail, though I realise this may make it longer than it already is.

On reading the paper I was keen to get using this and test the examples. I am not a Julia user, but have a reasonable level of skill with programming and modelling. I started with the GitHub repository and the associated documentation.
https://jianghuidu.github.io/SedTrace.jl/dev/guide/

I was able to install Julia, and follow the installation instructions to add the SedTrace package. I realise it is not the job of the author to teach users basics of Julia, but I somewhat embarrassingly was then unsure of how to get to the next step. I am not exactly sure where the installation put the git repo or how to run a Julia script. Whilst you have the workflow section I was not clear on what to type at the prompt. After an hour of training myself on Julia I decide to use the Visual Studio Code Julia plugin, and then click "Run" on the main script. I'm running the phbb example: main.pHBB1991.jl.  This gave a few package issues, which I was able to resolve as:

```
import Pkg; Pkg.add("NonlinearSolve")
import Pkg; Pkg.add("DataFrames")
```

then many other errors, meant that the only step that complete was creating "model_parameter_template.pHBB1991.uniform.xlsx". At that point I got the error:

```
Fatal error:
ERROR: MethodError: Cannot `convert` an object of type
JLD2.ReconstructedTypes.var"##Base.InvasiveLinkedList{Task}#332" to an
object of type Base.IntrusiveLinkedList{Task}

Closest candidates are:
  convert(::Type{T}, ::T) where T
   @ Base Base.jl:64
```

This experience was on a Mac Silicon 13.4 Julia 1.9.1.

I am sure I have just done something obvious wrong as a non-Julia (and Mac!) person. However, again in the spirit of making this type of modelling easier, I would really benefit

from another section in the documentation, which gives exact steps and allows a new user to get running the examples using only the downloaded GitHub example repo.

References:

Paraska, D.W., Hipsey, M.R. and Salmon, S.U., 2014. Sediment diagenesis models: Review of approaches, challenges and opportunities. *Environmental modelling & software*, *61*, pp.297-325.

---

## Author Comment (AC1)

**Response to Referee 1**

"SedTrace 1.0: a Julia-based framework for generating and running reactive-transport models of marine sediment diagenesis specializing in trace elements and isotopes", by Jianghui Du

The author provided a Julia based code for generating diagenetic reactive transport codes. The goal of the model development is well described and the code realizes the flexibility to enable various usages targeting at different sediment environments with different tracers of interest. The model's validity has been well examined and example simulations demonstrate the capacity of the model well. Accordingly, I think the paper is well-suited for publication in GMD. Followings are my comments on specifics that I hope the author may find of some use to improve the manuscript.

Reply: Thanks for your comments and suggestions!

Potential internal inconsistency. pH calculation in Sect. 4.2 does not seem to take into account detailed aqueous speciation (e.g., no accounts of Fe + CO3, Fe + HCO3, Fe +HS in Eqs. 27, 28, etc.). However, those are included in aqueous speciation calculation in Sect. 4.3. As long as concs. of included aqueous species with EIs or other tracked aqueous species are insignificant compared to total concs. of EIs/other tracked species, pH calculation as well as mass balance must not be significantly affected, but we can think of a situation where some ion pairs matter relative to EIs/other tracked species, e.g., ferruginous oceans in the Precambrian or deep depths of sediments where porewater chemistry is in equilibrium with some soluble solid phases. This may be my misunderstanding but potential problem here is model's internal inconsistency between aqueous speciation and pH modeling, and potential violation of charge balance in porewater. If this is the case, it should be clarified.

Reply: In SedTrace v1 the major ion speciation is fixed following that of modern seawater, and trace element speciation is assumed to have no impact on major ion speciation (such as HCO3 and CO3) because of low concentration. Thus, the current version of SedTrace is not suitable for cases when seawater/pore water composition is dramatically different from the modern seawater. In the revision, we will clarify this point.

Numerical diffusion. Especially related to Sect. 4.4. It is well known that numerical diffusion affects proxy record and thus different models have attempted to deal with it together with diagenetic reactions (e.g., Kanzaki et al., 2021; Munhoven, 2021). It probably does not affect steady state age-profiles, but the author should make it clear how the model deals with numerical diffusion and its effects on proxy reading or comparison with observations.

Reply: In advection dominated regime transient simulation will perform worse than steady-state simulation because of numerical diffusion using the finite volume complete flux (FVCF) scheme; regardless, even with numerical diffusion the FVCF scheme will ensure at least first-order convergence (ten Thije Boonkkamp and Anthonissen, 2010; ten Thije Boonkkamp and Anthonissen, 2011). As such, SedTrace v1 is best suited for steady-state simulations. That's why currently we do not include any examples of transient simulations of paleo-proxies. We will clarify this limitation of SedTrace in the manuscript.

L419-425. The author should show a plot of Logistic and Heaviside functions as a function of omega so that it is easy for the reader to compare the two functions and evaluate the approximation here as I have not seen this approximation elsewhere. Also, Eq. 24 does not look the same as that in L428. Please make sure they are consistent. Also, it would be better to show how this approximation affect the results where omega calculation can affect sediment profiles (e.g., Fig. 4).

Reply: Yes, there's a typo in Eq. 24. It should be $\tanh(\tau(\Omega-1))/2+1/2$. Thanks for pointing this out. Approximating Heaviside or other discontinuous functions by differentiable functions is common in other fields like engineering and deep learning (Iliev et al., 2017). In the following figure, we show the Heaviside function together with its Logistic approximation. As the parameter $\tau$ increases, the approximation becomes better.

[Figure]

We have compared the Santa Barbara Basin model results using the Heaviside function vs. the Logistic approximation ($\tau$=1000) and found the differences are negligible. For example, the relative difference of modeled CaCO3 concentration, mainly controlled by dissolution/precipitation reactions, is at most $10^{-7}$. The Logistic approximation is optional and the user can disable it by setting `AllowDiscontinuity = true` in `ModelConfig` when generating code. We will clarify this point in the revision.

Technical comments:

Code. I have tried to install and use the code to test example simulations in this paper (from examples directories) but ended up not being able to (failed with windows Cygwin and virtual Ubuntu in windows). This could be only because of me not used so much to Julia but it might be better to indicate under what environments the code has been tested so far and is supposed to work. This does not have to be in the text but anywhere else like Supplement or docs in code repository.

Reply: SedTrace was developed and tested on Windows 10. And using the GitHub action workflow we have tested it on the latest Ubuntu and macOS (non-Apple silicon chips) platforms. However, it has not been tested on Windows-Cygwin or -Linux virtual machines which we have no access. We did notice in the recent testing that a dependent package has had a breaking change after the release of SedTrace v1.0, causing code failure in certain cases. We have fixed this issue and released SedTrace v1.2 with improved version control system. We will continuously maintain and update SedTrace. And code issues can be reported to the Github page. We will clarify this in the code repository.

L106. Why does it have to be in mathematica notebook, which is not open to everyone? It would be better to give some final form of derivation in the main text or even supplement (or any form available to the reader) rather than saying that the reader can check them if they can use mathematica.

Reply: We have converted it to a PDF file now.

L106. maybe --> may be

Reply: Corrected.

L831. Sensitivity --> sensitive

Reply: Corrected.

L1097. It would be easier to read results if the author also provides delta values (1000 x ln alpha) in per mil.

Reply: Will do.

---

## Author Comment (AC2)

**Reply to Referee #2**

It was a pleasure to read through and learn about the new model Sed Trace, in the paper "**SedTrace 1.0: a Julia-based framework for generating and running reactive-transport models of marine sediment diagenesis specializing in trace elements and isotopes**" by Jianghui Du.

Reply: Thanks for your comments and suggestions!

The model being described is a 1D (pseudo-)steady-state implementation of a sediment early diagenesis model, that is able to include carbon and nutrients, trace elements, and isotopes. This type of model is notoriously difficult to setup, apply and master as users must have a good command of numerical methods, and many sub-disciplines of (computational) geochemistry, and experience with approaches to validate and assess. Yet, the model scope is broadly relevant to many topical areas of geoscience, and developing new methods and tools to reduce the barrier to entry is of great interest to the community. The paragraph starting line 62 may benefit from the paper by Paraska et al (2014) which similarly highlights the challenge after doing a meta-analysis of models.

Reply: We will include this reference for better contextualization.

As suggested in the title, the software developed in this case is quite unique, as the user can setup a tailored simulation relatively simply in an xlsx spreadsheet, and then this information is parsed to create full-featured simulation in the Julia language. Outputs are similarly written to xlsx for subsequent plotting/analysis. As someone who has spent much time wrestling with fortran versions of similar models (including CANDI), I found the idea quite clever and from a user experience point of view the approach of having a lot of complex configuration information nicely summarised in a xls spreadsheet was a very clean and flexible way to do this.

At the heart of the model is an implementation of a reactive-transport code following the traditional approaches introduced in the foundational models of CANDI, STEADYSED etc, and the numerical methods for discretisation and ODE solution seem closest to CANDI, though updated (e.g. using the robust CVODE). In my view, it is elegantly described and I found it hard to fault, with care and attention to detail in the equations.

The model allows for a grid transformation (e.g. changing resolution over depth). The opening of section 3.1 may benefit from an improved opening/contextual statement setting the scene for this; whilst experienced users will get this concept, it might be confusing new users unclear what they should do or consider when making a grid and options available.

Reply: We will add a statement on the use of grid transformation, e.g., capturing sharp chemical gradients at certain locations.

Also, our experience with CANDI numerical code was that user's need to be aware of setting the grid parameters, and in some cases they would lose some mass conservation – this was albeit in more unsteady conditions, but this made me wonder if mass balance testing had been done on this code, or could be reported to the user to check for grid issues?

Reply: CANDI uses a finite difference method which cannot guarantee mass conservation. That's why we have chosen a finite volume method for SedTrace which is conservative. We will reiterate this point in the revision.

The model includes options for both kinetic and equilibrium chemical reactions, and is quite flexible in allowing users to formulate their reactions and their stoichiometry. The parser is able to allow users to easily tailor their settings and reaction network in Excel rather than having to edit code, and includes options to check stoichiometric balance. The approach to allow for Omega on reactions allows for precipitation/dissolution, and the code adopts a logistic rather than Heaviside style to smooth the numerical solution when quantities are close to equilibrium.

The pH solution is also well implemented, essentially conserving charge by adopting the concept of the equilibrium invariant concentration (similar to a component definition in PHREEQC). This can be

a headache when dealing with boundary conditions, but this is addressed in the model. A comment here how this relates to or is different from traditionally used approaches may be warranted.

Reply: Hofmann et al (2008) has given a great summary of the different approaches of pH modeling which we will make reference to. SedTrace follows the direct substitution approach of Hofmann et al (2008) which differs from other approaches in that pH is modeled dynamically so the impact of reaction and transport on pH can be easily partitioned. We will emphasize this contrast with traditional approaches in the revision.

Overall, I found the model description excellent, and the software is full-featured and should be published. My main comment is that there is a fair amount of assumed knowledge required to digest these options and methods, and in the spirit of making this type of modelling easier for a wider audience, then the paper may also benefit from some careful additions of some leading or contextualising sentences before diving into some of the detail, though I realise this may make it longer than it already is.

Reply: Thanks for this suggestion. We will add a few more explanations of how our approach differs from other models and the rationales behind in the revision, especially regarding the numerical discretization, pH and speciation modeling.

On reading the paper I was keen to get using this and test the examples. I am not a Julia user, but have a reasonable level of skill with programming and modelling. I started with the GitHub repository and the associated documentation. https://jianghuidu.github.io/SedTrace.jl/dev/guide/

I was able to install Julia, and follow the installation instructions to add the SedTrace package. I realise it is not the job of the author to teach users basics of Julia, but I somewhat embarrassingly was then unsure of how to get to the next step. I am not exactly sure where the installation put the git repo or how to run a Julia script. Whilst you have the workflow section I was not clear on what to type at the prompt. After an hour of training myself on Julia I decide to use the Visual Studio Code Julia plugin, and then click "Run" on the main script. I'm running the phbb example: main.pHBB1991.jl. This gave a few package issues, which I was able to resolve as:

import Pkg; Pkg.add("NonlinearSolve") import Pkg; Pkg.add("DataFrames")

then many other errors, meant that the only step that complete was creating "model_parameter_template.pHBB1991.uniform.xlsx". At that point I got the error:

Fatal error:

ERROR: MethodError: Cannot `convert` an object of type JLD2.ReconstructedTypes.var"##Base.InvasiveLinkedList{Task}#332" to an object of type Base.IntrusiveLinkedList{Task}

Closest candidates are: convert(::Type{T}, ::T) where T

@ Base Base.jl:64

This experience was on a Mac Silicon 13.4 Julia 1.9.1.

I am sure I have just done something obvious wrong as a non-Julia (and Mac!) person. However, again in the spirit of making this type of modelling easier, I would really benefit from another section in the documentation, which gives exact steps and allows a new user to get running the examples using only the downloaded GitHub example repo.

Reply: We are sorry for this problem. This error was caused by a breaking change in the dependent package JLD2.jl after updating to Julia v1.9.0 or higher. SedTrace v1.0 didn't have a good version control system and that's why when the reviewer installed it the newer versions of dependent packages were called which were different from the versions when SedTrace v1.0 was released. Now we have used the Julia CompatHelper.jl and GitHub tools for better version control. This will make sure the correct versions of dependent packages are used in the future. These improvements have been included in the new release SedTrace v1.2. We have also modified the "Installation" section and added a "First example" section in the documentation to show the user how to run the examples

discussed in the manuscript.

References:

Paraska, D.W., Hipsey, M.R. and Salmon, S.U., 2014. Sediment diagenesis models: Review of approaches, challenges and opportuniFes. *Environmental modelling & so1ware*, *61*, pp.297-325.